# Genetic and dietary determinants of gut microbiome-bile acid interactions in the BXD genetic reference population

Xiaoxu Li [1], Alessia Perino [2], Jonathan Sulc[1], Antoine Jalil[2], Giacomo V. G. von Alvensleben[1], Jean-David Morel [1], Qi Wang [1], Alexis Rapin [1], Hao Li [1,3], Kristina Schoonjans [2] ✉ & Johan Auwerx [1] ✉

The gut microbiome is crucial in regulating overall physiology and communicates with the host through various microbial-derived metabolites, including secondary bile acids (BAs). However, mechanisms underlying the gut microbiome-BA crosstalk (gMxB) are still poorly understood. Here, we assess the postprandial cecal microbiome, BA levels, and colon transcriptome of male BXD mice fed with a chow or high-fat diet, and find that genetic and dietary factors shift microbiome composition and affect gMxB. Four diet-dependent co-mapping genetic loci associated with gMxB, including the interaction between *Turicibacter sanguinis* - plasma cholic acid, are identified using systems genetics approaches. By integrating human MiBioGen database, we prioritize *PTGR1* and *PTPRD* as candidate genes potentially regulating identified gMxB. The human relevance of these candidates on metabolic health is investigated using data from the UK biobank, FinnGen, and million veteran program databases. Overall, this study illustrates potential modulators regulating gMxB and provides insights into gut microbiome-host communication.

In the last decade, the gastrointestinal tract has been increasingly recognized as a complex signaling organ communicating with multiple tissues, including the brain and other metabolic organs[1,2]. Increasing evidence indicates that the dysregulation of the gut microbiome-host communication network contributes to the development of obesity and its co-morbidities (e.g., type 2 diabetes)[3,4] through various microbially-derived metabolites, including secondary bile acids (BAs)[5]. BAs are a class of structurally diverse cholesterol derivatives. In addition to their role as digestive surfactants, they also function as versatile signaling molecules mediating adaptive responses to nutritional cues[6]. Primary BAs are produced in the liver, conjugated with the amino acids glycine or taurine, stored in the gallbladder, and secreted into the intestine after food intake to facilitate the absorption of lipids and lipid-soluble vitamins of dietary origin[7]. In the intestine, the gut microbiome can deconjugate primary BAs, and then metabolize them to secondary BAs through dehydroxylation, dehydrogenation and epimerization[5]. In turn, BAs modulate the gut microbiome composition by exerting bacteriostatic and bactericidal effects on luminal bacteria through their detergent action or indirectly through activation of BA receptors[5].

The interdependence between the gut microbiome and BAs, hereafter termed as the gut microbiome-BA crosstalk (gMxB), can be influenced by both genetic and environmental factors[8,9], and plays important roles in host metabolic homeostasis[6,10,11]. In mice, chronic fat intake is known to reduce the complexity of the microbiome[12,13], comparable to what is observed in human subjects with obesity[14,15]. The microbiome is capable of transmitting metabolic changes of obesity, as transfer of gut microbiome from overweight mice increases

[1]Laboratory of Integrative Systems Physiology, Institute of Bioengineering, École Polytechnique Fédérale de Lausanne, Lausanne, Switzerland. [2]Laboratory of Metabolic Signaling, Institute of Bioengineering, École Polytechnique Fédérale de Lausanne, Lausanne, Switzerland. [3]Center for Mitochondrial Biology and Medicine, The Key Laboratory of Biomedical Information Engineering of Ministry of Education, School of Life Science and Technology, Xi'an Jiaotong University, Xi'an, China. ✉e-mail: kristina.schoonjans@epfl.ch; admin.auwerx@epfl.ch

weight gain in gnotobiotic animals[16,17]. Chronic high-fat diet (HFD) feeding can also shift the BA composition in different tissues[9,18,19]. Modulation of BA composition, like decreasing the proportion of non-12-hydroxylated (non-12-OH) BAs, also contributes to the susceptibility to obesity[20,21]. Changes in gMxB can thus have crucial effects on host physiology and metabolic health[22,23]. Despite these important observations, mechanisms underlying the crosstalk between gut microbiome and host metabolism as well as the role of BAs in this interaction, remain poorly understood. Revealing the effects of genetic, environment (e.g., diet), and gene-by-environment interaction (GxE) on gMxB is critical for understanding intestinal and whole-body homeostasis.

Studying GxE effects in humans is challenging because of the heterogeneities in diet and lifestyle. This can be complemented using mouse models, allowing to control several environmental factors such as temperature and diet, that impact gut physiology, microbiome homeostasis and metabolic responses[9,24,25]. Additionally, mouse studies allow to systematic access to deep organs, to elucidate tissue-specific mechanisms[24,25]. However, the characterization of the microbiome, BAs and their interaction in mice has been generally restricted to animals on a single genetic background[25]. Since host genetic diversity is an important determinant of the gut microbiome[26–29], experiments on a single genetic background model limit their ability to model the factors that govern the gut microbiome in human populations. Mouse genetic reference populations (GRPs) mimicking the genetic heterogeneity of human populations are used to study complex traits using systems genetics approaches[24,25]. For example, quantitative trait locus (QTL) mapping was used to investigate the genetic effects on the gut microbiome in the BXD GRP[27,30] (derived from a cross between the C57BL/6J and DBA/2J strains) in a fasted state. In addition, we previously designed a study in fed BXD strains that allowed to investigate how genetics and diet impact BA homeostasis[9]. In this study, we profiled postprandial BA levels in feces, liver, and plasma, and found a number of host genetic and dietary factors affecting BA homeostasis, but did not address the interaction between gut microbiome and BA levels[9].

In this follow-up study, we characterized the cecal microbiome using 16S ribosomal RNA sequencing and analyzed the microarray-based host transcriptome of the proximal colon from 32 BXD strains fed with a chow diet (CD) or HFD (~5 male mice per strain and diet). By integrating these new cecal microbiome profiles with BA data profiled on the same mice in our previous study[9], we show that the effects of HFD on gut microbiome are strain-specific and that gut microbiome-host physiology crosstalk is diet-dependent. This forms the foundation of our analysis, which characterizes the genetic and dietary factors influencing these interactions. Multi-omics datasets were also integrated using systems genetics approaches, identifying four diet-dependent genetic loci that are associated with distinct gMxB, including established (e.g., *Turicibacter sanguinis* (*T. sanguinis*) - plasma cholic acid (CA)) and unknown gMxB interactions (e.g., *Bacteroides uniformis* (*B. uniformis*) - fecal taurolithocholic acid (TLCA)). Candidate genes in these loci were shortlisted using systems genetics prioritization in BXD mice and further prioritized using human gut microbiome genetic associations and putative causal effects inferred using Mendelian randomization (MR). These analyses led to the identification of two candidate genes regulating gMxB – *PTPRD* and *PTGR1*. Taken together, our findings provide insights into the understanding of gut microbiome-BA associations and their potential effects on metabolic health.

## Results

### The microbiome composition is determined by GxE interactions in BXDs

In this study, we leveraged a male BXD mouse panel of 32 strains fed with a CD or HFD from 8 to 29 weeks of age to explore the genetic and dietary determinants of the gMxB. As described previously[9], BAs were profiled in plasma (20 weeks), feces (24 weeks), and liver (29 weeks). Plasma samples were collected before (T0), and 30 (T30) or 60 minutes (T60) after controlled test meal gavage feeding. These mice were euthanized in a controlled postprandial state (Methods) and multiple organs were collected for investigation (Fig. 1a). As expected, HFD feeding significantly increased the body weight of the mice and decreased cecum weight (Fig. 1b). To elucidate the dietary effects on the gut microbiome, we first performed principal coordinate analysis and found that mouse cecal microbiome profiles were significantly separated by diet (Fig. 1c), suggesting that 21 weeks of HFD feeding shifted the cecal microbiome composition in BXD mice. In line with previous studies[31,32], the alpha diversity–a measure of the microbial diversity within an ecological community–was significantly decreased upon HFD, whether we consider the richness (the number of observed bacteria within a sample), evenness (the distribution of bacteria within a sample), or Shannon index (the bacteria diversity within a sample considering both richness and evenness) (Fig. 1d). Most identified bacteria belonged to eight phyla, among which Firmicutes (also known as Bacillota) was the most abundant, followed by Bacteroidota (also known as Bacteroidetes), in both CD and HFD (Fig. 1e).

Changes in gut microbiome in response to HFD were strain-specific (Fig. 1e). Specifically, the ratio of Firmicutes to Bacteroidota (F/B ratio) was significantly changed upon HFD in two parental strains, increasing in the C57BL/6J[31,33], but decreasing in the DBA/2J. As expected, the change in the F/B ratio upon HFD varied in the BXD strains due to their distinct genetic backgrounds inherited from the two parental strains (Fig. 1e, Supplementary Fig. 1a). Similarly, the Shannon index was significantly decreased upon HFD in most BXD strains, but not all (e.g., BXD6 and BXD11) (Supplementary Fig. 1b). We also found that the Bray–Curtis dissimilarity between individuals of a given strain was significantly smaller than that between strains (tested within each diet separately) (Supplementary Fig. 1c), suggesting the profound genetic effects on microbiome composition. Overall, these results illustrated not only the HFD effects on the gut microbiome, but also highlighted the important roles of genetic background and GxE, enabling us to investigate genetic factors that regulate the gut microbiome diversity in response to dietary challenges.

### Effects of HFD feeding on the genus-level cecal microbiome and colon transcriptome

To further investigate changes in bacterial abundance upon HFD, we first performed a population-level analysis of composition of microbiomes (ANCOM) at the genus level and identified 15 differentially abundant genera (DAGs) with absolute $\log_e$(Fold change) > 1 and q value < 0.05 (Fig. 2a). Among these 15 DAGs, 10 belonged to the Firmicutes phylum. Of note, *Lactococcus* was the most increased genus upon HFD, while *Turicibacter* was significantly decreased, and barely detected in HFD. *Bifidobacterium* is another genus that was significantly decreased upon HFD. The changes of these genera upon HFD were consistent with previous reports[31,34,35]. We then performed ANCOM for each mouse strain and found that the bacterial changes upon HFD in most, but not all, BXD strains were concordant although heterogeneous in amplitude (Fig. 2b), suggesting the impacts of GxE effects on the gut microbiome composition. For example, the abundance of *Anaerotignum* was increased upon HFD in BXD8, whereas it was decreased or not detected in the other BXD strains (Fig. 2b).

To explore the effect of HFD on host gene expression and its interaction with the microbiome, we first profiled the colon transcriptome and assessed the genes altered upon HFD. While the transcriptomes of HFD- and CD-fed mice did not separate well from each other in the principal component analysis (Supplementary Fig. 2a), we identified similar population-level differentially expressed genes to those reported in another BXD study on HFD-fed mice, though that study analyzed samples collected during the fasted state[36]. For

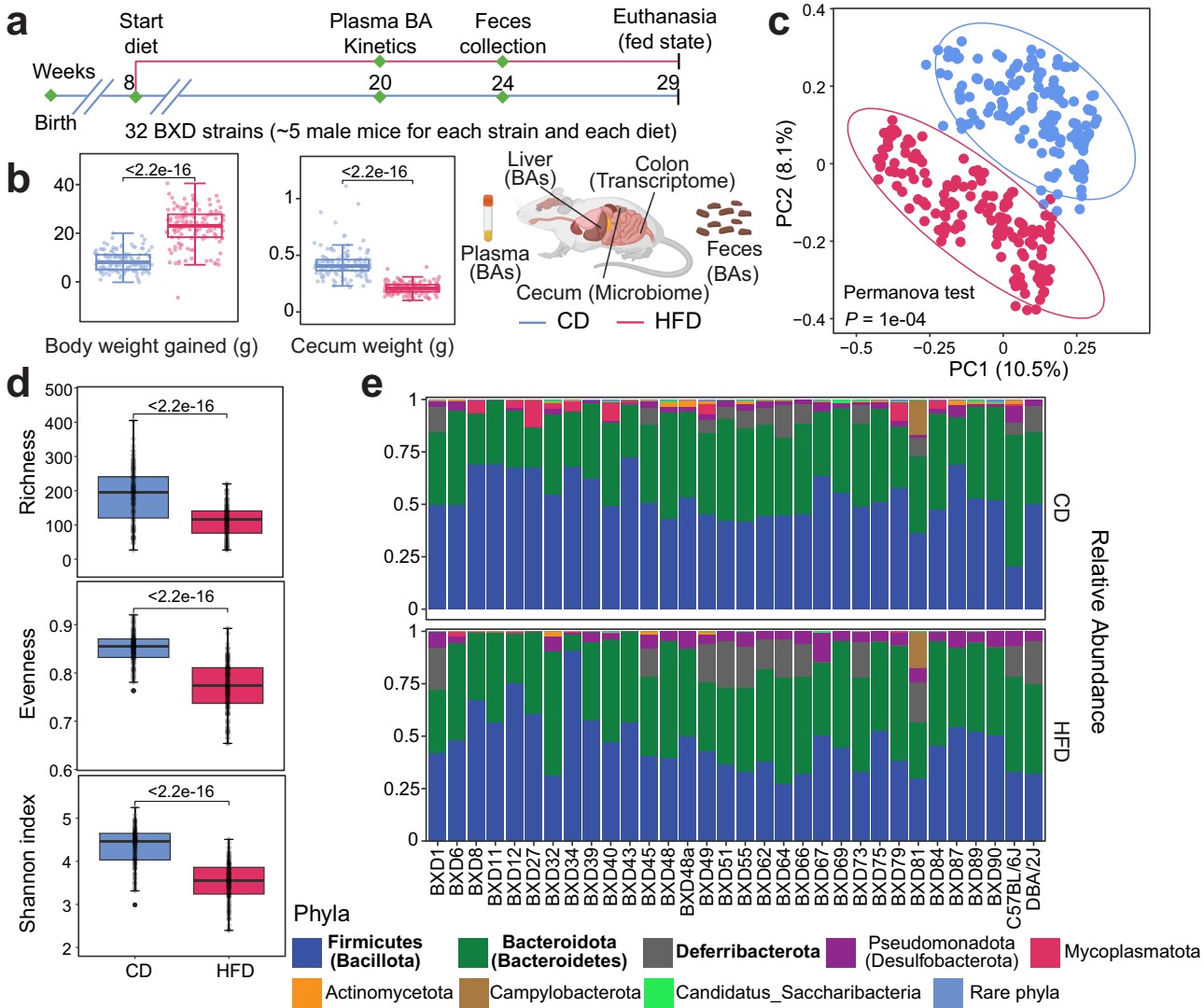

**Fig. 1 | Effects of HFD feeding on the composition of the cecal microbiome in the BXD mouse genetic reference population. a** Schematic of the experimental pipeline of the postprandial BXD study[9]. Plasma and fecal samples were collected at 20 and 24 weeks of age, respectively, for bile acid (BA) measurements. At 29 weeks, the mice were euthanized, and the liver, colon, and cecum were harvested for BA analysis, transcriptome profiling, and microbiome analysis, respectively. Icons are created in BioRender. Perino, A. (2025) https://BioRender.com/xdy313m. **b** Boxplots showing the body weight gained from 8 to 29 weeks of age (N = 130 [CD] and 129 [HFD]), and cecum weight (N = 132 [CD] and 135 [HFD]) measured at the end of the study. Chow diet: CD, high-fat diet: HFD. P values were calculated by two-

tailed Student's t-test. **c** Principal coordinate analysis (PCoA) of cecal microbiome profiles based on the Bray–Curtis dissimilarity under CD and HFD. P value was calculated by permutational multivariate analysis of variance (PERMANOVA) performed with 9999 permutations. **d** Boxplots of richness, evenness and Shannon index of BXD gut microbiome in CD (N = 144) and HFD (N = 151). P values were calculated by two-tailed Student's t-test. **e** Relative abundance of the indicated bacteria phyla across strains and diets. Phyla were indicated by color and the three most enriched phyla in cecum were indicated in bold. Source data are provided as a Source Data file.

example, serum amyloid A 1 and 3 (*Saa1* and *Saa3*), which are involved in the inflammatory response[37], were up-regulated upon HFD. In contrast, the cytochrome P450, family 2, subfamily c, polypeptide 55 (*Cyp2c55*) that inhibits colon tumorigenesis[38], and Carboxylesterase 2 (*Ces2a*), which is decreased in experimental colitis models[39], were down-regulated (Supplementary Fig. 2b). This suggests that the effects of HFD on colon gene expression are robust and remain consistent despite short-term changes in feeding status. We next sought to understand the association between host colon gene expression and the gut microbiome, which is critical to the development of intestinal disorders[40], therefore, we investigated the association between transcriptome and microbiome and assessed the impact of dietary challenge on their interaction. We performed Procrustes analysis using paired data in each diet and observed a significant concordance between host gene expression and gut microbiome composition

across BXD strains under CD (P = 0.047, Fig. 2c), but not HFD (P = 0.31, Fig. 2d), suggesting that HFD may disrupt the overall host gene-microbiome communication.

## Uncovering HFD-specific host gene-gut microbiome associations

Although there is no overall association between host colon gene expression and the gut microbiome under HFD, specific groups of gut bacteria and host genes can still be associated. To uncover the HFD effects on the group-level associations between the colon transcriptome and the gut microbiome, we applied sparse canonical correlation analysis (CCA)[41,42], a machine learning-based approach that has been used in multi-omics integration[42] (e.g., gene expression and microbiome datasets[40]). This method identifies subsets of bacteria and host genes (termed components (C)) such that a linear

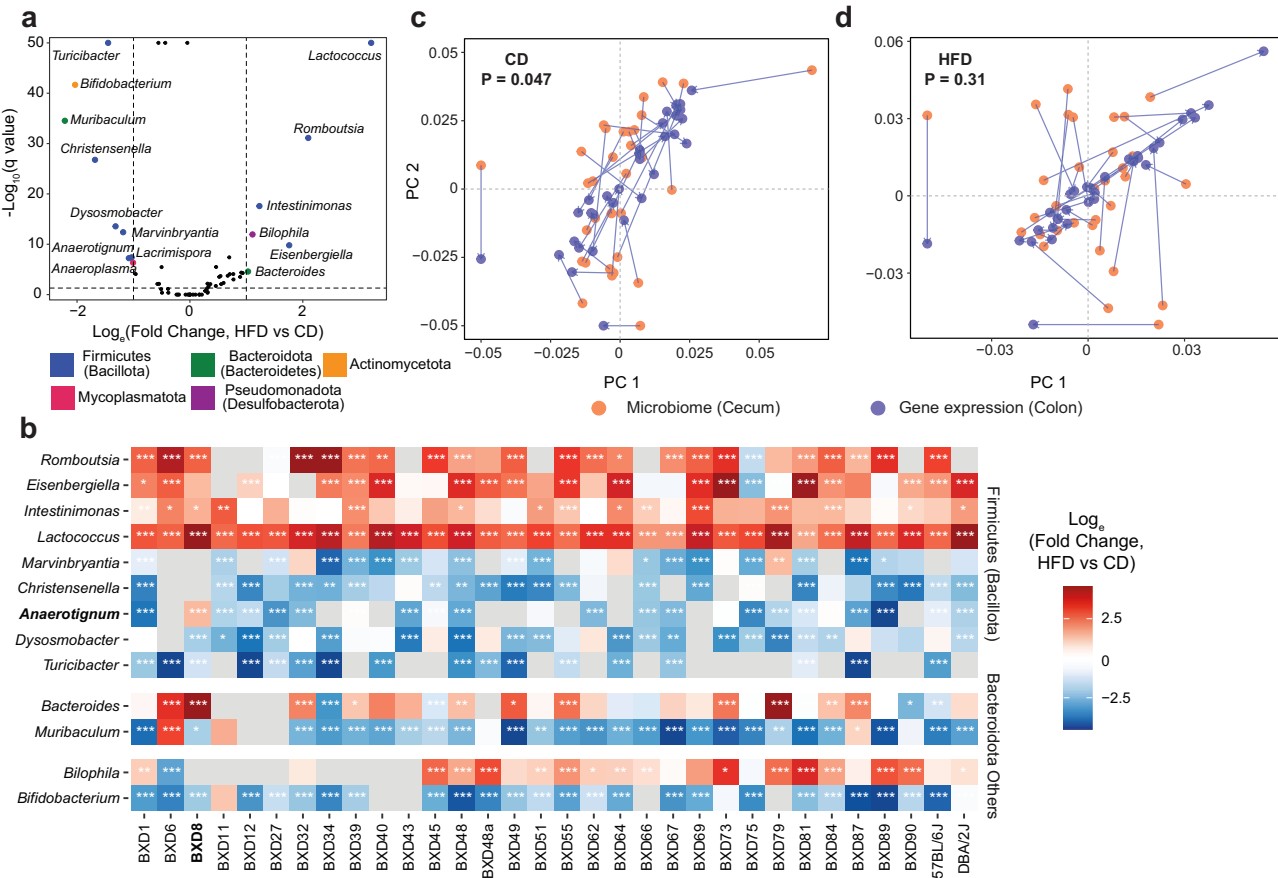

**Fig. 2 | HFD feeding alters the gut microbiome at the genus level. a** Volcano plot showing the effect of HFD on the abundance of each bacterial genus across all BXD mouse strains. Genera with $\log_e$(Fold change) > 1 and $q$ < 0.05, obtained from analysis of composition of microbiomes – ANCOM, were identified as differentially abundant genera (DAGs) and colored by the corresponding phylum. **b** Heatmap showing a subset of genera that were significantly changed ($q$ < 0.05, ANCOM) upon HFD in both diet-independent view and in most BXD strains (more than 15). $q$ values are indicated as follows: *$q$ < 0.05; **$q$ < 0.01; ***$q$ < 0.001. Missing values in both diets were indicated by gray color. Procrustes analyses indicating the overall association between variation in host colon gene expression and gut microbiome composition under CD (**c**) and HFD (**d**) based on the Aitchison's distance of host gene expression data (response) and Bray-Curtis distance of gut microbiome data (predictor). P: *P* value were calculated in Procrustes analyses. Source data are provided as a Source Data file.

combination of one is maximally correlated with the other (Fig. 3a). To further explore the biological relevance of these associations, we then performed over-representation analysis (ORA) on host genes within each component (Fig. 3a). Among the sparse CCA analysis, only two components were significantly correlated in CD (Fig. 3b), while in HFD, four significantly correlated components were identified (Benjamini–Hochberg (BH)-adjusted *P* < 0.05) (Fig. 3c).

The three bacterial species with the highest weights within CD component 1 (C1 (CD)), include *Faecalibaculum rodentium* (*F. rodentium* - CCA coefficient: −0.43), *Turicibacter sanguinis* (*T. sanguinis* - CCA coefficient: −0.42), and *Lactobacillus taiwanensis* (*L. taiwanensis* - CCA coefficient: −0.40) (Fig. 3b). *F. rodentium* has been shown to modulate cell growth[43], and indeed, C1 (CD) was related to genes involved in cell proliferation, mitochondrial function and cellular stress, notably MYC targets, respiratory electron transport, oxidative phosphorylation (OXPHOS), and heat shock factor 1 (HSF1) targets (Fig. 3d). Under HFD, bacterial species in the C3 (HFD) were not only significantly related to cell proliferation-and mitochondria-related genes, but also associated with genes involved in immune system, while those in the C7 (HFD) were only suggestively associated with inflammatory responses (Fig. 3e). Of note, some bacteria can interact with distinct bacterial species and host genes under altered dietary conditions, thereby participating in different biological processes[40]. For example, *Bacteroides acidifaciens* and *L. taiwanensis*, are related to genes involved in cell proliferation under CD and inflammation under

HFD (Fig. 3d, e). We also found that bacteria groups associated with inflammatory response were only identified in HFD, suggesting that specific bacteria are either a cause or a consequence of host intestinal inflammation upon HFD feeding.

## HFD feeding reshapes the associations between gut microbiome and BAs

Emerging evidence has shown that the gut microbiome plays a key role in metabolic homeostasis and its-associated disorders by modulating BA homeostasis[18,23,44]. The ratio of microbially-derived secondary BAs to host-derived primary BAs provides insights into the gut microbiome's influence on the BA pool. In CD, a positive correlation was found between the Shannon index and the secondary BA/primary BA ratio in the feces, liver, and plasma (60 minutes after a test meal gavage, T60), and these correlations were lost upon HFD feeding (Fig. 4a). This may be due to HFD-induced reductions in the alpha diversity of gut microbiome[31,32] (Fig. 1d), ultimately disrupting microbiome-BA interactions (Fig. 4a). Likewise, we found several diet- and biological compartment-dependent correlations between bacterial species and the secondary BA/primary BA ratio. Specifically, *Muribaculum intestinale, Lactobacillus intestinalis, Bifidobacterium globosum* and *Muribaculum gordoncanteri* were only positively correlated with secondary BA/primary BA ratio across all analyzed biological compartments in CD-fed mice (Fig. 4a). *Schaedlerella arabinosiphila* was found to be positively correlated with the secondary BA/primary

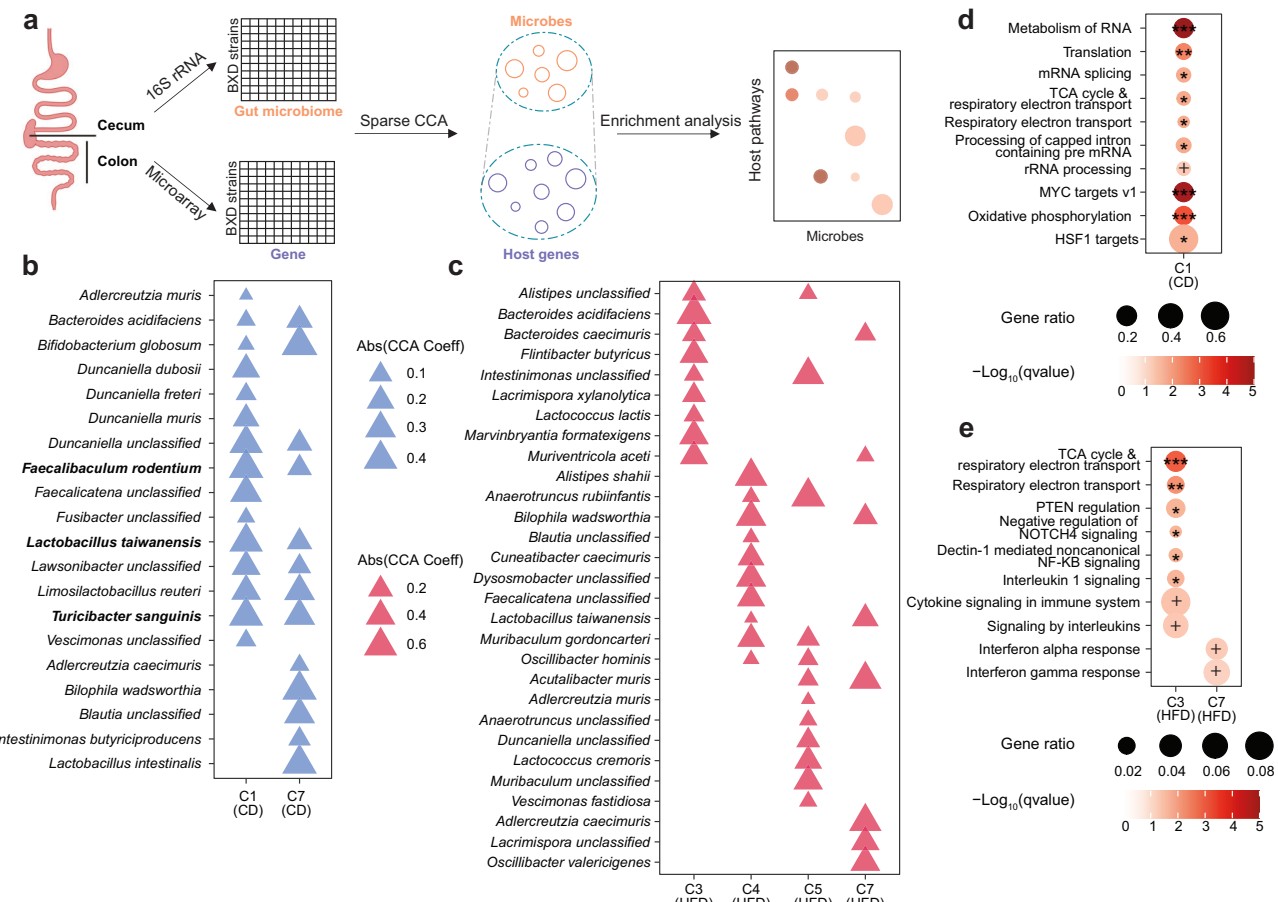

**Fig. 3 | Immunoregulatory pathways are specifically associated with HFD-induced changes in the cecal microbiome. a** A pipeline for integrating gut microbiome and host colon gene expression to explore microbiome composition-associated pathways. The gene set-bacterium group associations were identified using sparse canonical correlation analyses (CCA, Benjamini-Hochberg (BH)-adjusted $P < 0.05$) and enrichment analyses were applied to find bacteria group-associated pathways. Icons are created in BioRender. Perino, A. (2025) https://BioRender.com/wcqza6l. The composition of the bacteria groups within

components (e.g., C1 and C7, Benjamini-Hochberg (BH)-adjusted $P < 0.05$) identified using sparse CCA under CD (**b**) and HFD (**c**). The size of triangles indicates the absolute value of sparse CCA coefficients of bacteria. Over-representation analysis (ORA) indicating the host pathways enriched for genes associated with gut bacteria groups identified by sparse CCA under CD (**d**) and HFD (**e**). Pathways or defined genesets with a $q$ value less than 0.1 were shown and -$\log_{10}$(q value) were indicated by as follows: + $q < 0.1$; *$q < 0.05$; **$q < 0.01$; ***$q < 0.001$. Source data are provided as a Source Data file.

BA ratio in feces upon CD and in liver and T60 plasma upon HFD, showing diet-specific correlation patterns (Fig. 4a). Conversely, some bacteria showed consistent positive correlation across both diets, such as *Bilophila wadsworthia* in plasma (T30) and *Vescimonas fastidiosa* in plasma (T0 and T60) and liver (Fig. 4a). Notably, *Paramuribaculum intestinale* displayed an opposite trend, positively correlating with the secondary BA/primary BA ratio in CD but negatively in HFD (Fig. 4a). These findings suggest that these bacterial species are potentially associated with the balance between these functionally distinct BA classes, either regulating them or being influenced by them.

To further explore potential bacterium-BA links, we performed Lasso penalized regression and Spearman correlation analyses on the abundance of each bacterium and the relative abundance of each BA (expressed as a fraction of the total in each biological compartment: BA X /all BAs) (Fig. 4b). We fitted the Lasso penalized regression using the relative abundance of each BA as the response variable and the abundances of gut bacterial species as predictors. 41 and 37 bacterial species-BA associations were identified by stability selection under CD and HFD, respectively (Fig. 4c, d). Only a few associations were conserved across diets identified from Lasso penalized regression or Spearman correlation, including *V. fastidiosa* with liver glycocholic acid (GCA) and plasma T30 tauro-β-muricholic acid (TβMCA), *F. rodentium* with plasma T0 ursodeoxycholic acid (UDCA) and

*Christensenella hongkongensis* with plasma T60 THDCA. However, the majority of associations were diet-dependent, suggesting that HFD alters the bacterial species-BA relationships (Fig. 4c, d). Specifically, in CD-fed mice, *T. sanguinis* exhibited a consistent positive correlation with CA in plasma at all three time points (Fig. 4c). Additional diet-specific associations included *M. gordoncarteri*, which was positively linked with GCA and THDCA in the livers of CD fed mice (Fig. 4c). In contrast, negative associations under HFD were observed, such as *Bacteroides acidifaciens* with tauro-alpha-muricholic acid (TαMCA) in the liver and *V. fastidiosa* with TβMCA in plasma T30 (Fig. 4d). Overall, these findings provide insights into the gut microbiome-BA crosstalk and highlight unexplored potential bacterial species-BA associations that are influenced by dietary conditions.

## Dissecting the genetic loci regulating gMxB across diets

In addition to dietary effects, host genetics also play a crucial role in shaping gut microbiome composition and BA profiles[9,45]. Therefore, we investigated whether certain host genetic loci are related to both layers, potentially pointing to a shared regulatory mechanism. To identify genetic loci associated with bacterial abundance, we performed QTL mapping of the cecal microbiome at both the genus and species level. Under CD, we identified two QTL peaks located on chromosome (Chr) 4, which were related to the abundance of

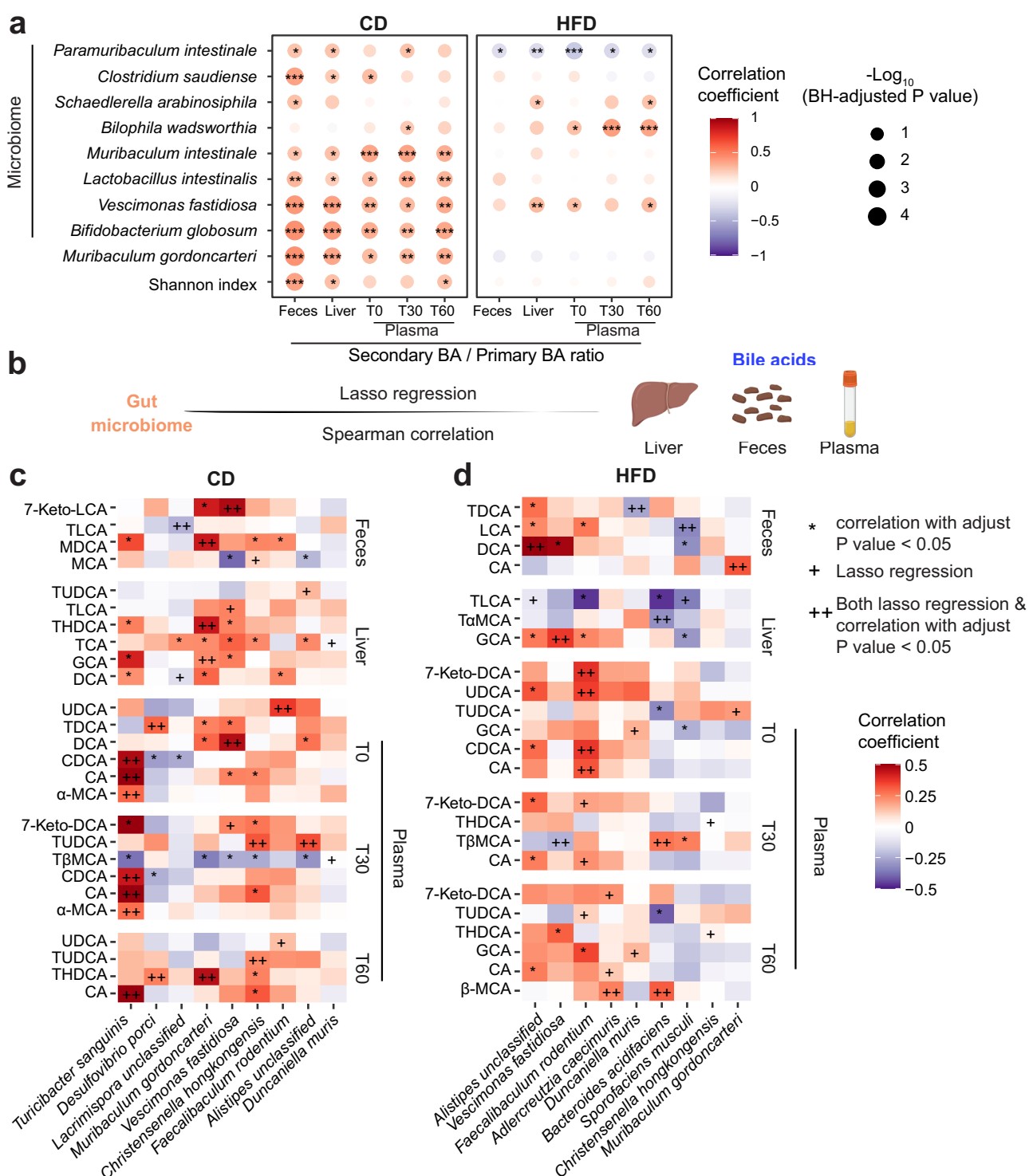

**Fig. 4 | Diet-dependent associations between the abundance of specific bacterial species and BAs. a** Heatmap showing the Spearman correlation between the ratio of secondary to primary BAs and gut bacterial species. Only bacterial species with significant correlations in at least three biological compartments or time points were shown. -Log$_{10}$(BH-adjusted $P$ value) is represented by dot size and indicated as follows: * BH-adjusted $P < 0.05$; ** BH-adjusted $P < 0.01$; *** BH-adjusted $P < 0.001$. **b** Schematic of cecal microbiome-relative abundance of each BA (expressed as a fraction of the total in each biological compartment: BA X /all BAs) correlation pipeline in three biological compartments. BAs were extracted from the plasma, feces, and liver. Lasso regression and Spearman correlation were applied. Icons are created in BioRender. Perino, A. (2025) https://BioRender.com/jyrxtgb. Heatmap of bacterial species-relative abundance of each BA pairs selected by just Lasso regression analysis (indicated with +), by just Spearman correlation analyses (indicated with *, BH-adjusted $P$ value < 0.05), or by both Lasso regression and Spearman correlation analyses (indicated with ++) under CD (**c**) and HFD (**d**). Bacterial species significantly associated with at least two BAs were shown. Color indicates the Spearman correlation coefficient. The abbreviations of BA are listed in the supplementary Table 1. Source data are provided as a Source Data file.

## Table 1 | Co-mapping QTL peaks across both bacterial abundance and BA profiles under CD and HFD, related to Fig. 5a–d

| QTL | Bacterium | Chr | Pos (Mb) | LOD | Start (Mb) | End (Mb) | Diet | Bile Acids |
|---|---|---|---|---|---|---|---|---|
| gMxB1 | *Turicibacter; T. sanguinis* | 4 | 62.65 | 4.26 | 46.35 | 68.27 | CD | Plasma: CA, CDCA, CA/all, TCA/all; Liver: CA/all |
| gMxB2 | *Turicibacter; T. sanguinis* | 4 | 74.25 | 3.77 | 68.27 | 88.04 | CD | Liver: CA/all |
| gMxB3 | *B. uniformis* | 7 | 46.0 | 3.42 | 40.9 | 46.4 | HFD | Feces: α-MCA, TLCA, UDCA |
| gMxB4 | *Bacteroides* | 7 | 56.47 | 3.44 | 55.65 | 66.32 | HFD | Feces: 7-keto-LCA, 7-keto-DCA/all |

*Chr* chromosome, *Pos* position, *LOD* logarithm of the odds, start and end regions are in Mbp.

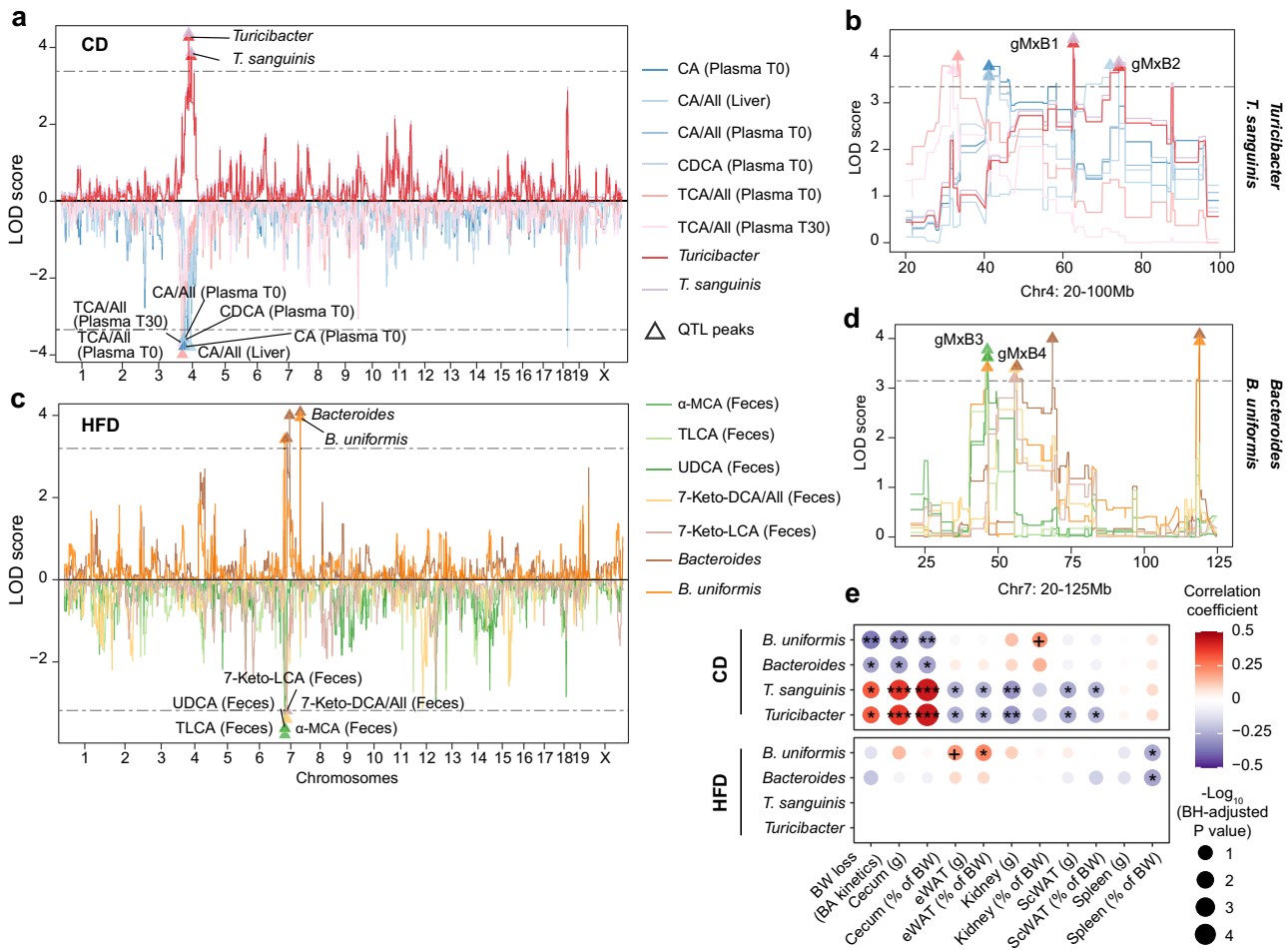

**Fig. 5 | QTL mapping identifies co-localized QTL peaks for bacteria and specific BA species under CD and HFD. a** Manhattan plot showing the co-localized QTL peaks for cecal microbiome and BA levels (abundance or ratio) under CD. **b** The overlapped QTL peaks of gut microbiome-BA interaction (gMxB) on chromosome 4 under CD. **c** Manhattan plot showing the colocalized QTL peaks for cecal microbiome and BA levels (abundance or ratio) under HFD. **d** The overlapped QTL peaks of gMxB on chromosome 7 under HFD. Dash lines indicate the minimum threshold of QTL mapping for indicated traits. **e** Dot plot showing the Spearman correlation between gut bacteria and metabolic traits in the BXD panel. The color indicates the correlation coefficient while the -Log₁₀(BH-adjusted *P* value) is represented by dot size and indicated as follows: + BH-adjusted *P* value < 0.1; * BH-adjusted *P* < 0.05; ** BH-adjusted *P* < 0.01; *** BH-adjusted *P* < 0.001. BW: body weight. ScWAT: subcutaneous white adipose tissue, and eWAT: epididymal white adipose tissue. Source data are provided as a Source Data file.

*Turicibacter* (Supplementary Fig. 3a), and one *Bacteroides*-related peak located on Chr 7 that was conserved in both diets (Supplementary Fig. 3a, b). These three QTL peaks at the genus level (Supplementary Fig. 3a, b) co-localized with QTL peaks for their relative species, *T. sanguinis* and *B. uniformis*, respectively (Supplementary Fig. 3c, d). Under HFD, we identified two additional QTL peaks for *Bacteroides* on Chr 7, and one for *Intestinimonas* on Chr 2 (Supplementary Fig. 3b). We also observed an additional QTL peak for *B. uniformis* on Chr 7, located near the three *Bacteroides* QTL peaks, though they did not overlap (Supplementary Fig. 3d).

In parallel, we performed QTL mapping for the absolute concentration and relative abundance (BA X/all BAs) of each BA across biological compartments and time points. By comparing the BA and microbiome QTL peaks (Table 1, Supplementary Table 2), we identified four gMxB QTL peaks that appear to relate to both bacterial genera and their respective species. Specifically, two shared genetic loci on Chr 4 were associated with the abundance of *Turicibacter* and, more specifically, *T. sanguinis*, as well as some liver and plasma BAs under CD (95% confidence intervals of these QTLs were overlapped). The first locus (gMxB1 QTL) associated with the CA/all ratio in plasma (T0) and

liver as well as CA, Chenodeoxycholic acid (CDCA), and the Tauro-cholic acid (TCA)/all ratio in plasma (T0 or T30) (Fig. 5a, b). The second locus (gMxB2 QTL) was only associated with CA relative abundance in the liver (Fig. 5a, b). These findings are consistent with the positive correlation between *T. sanguinis* abundance and total CA and CDCA in plasma (Fig. 4c). Under HFD, another shared genetic locus (gMxB3 QTL) on Chr 7 was associated with the abundance of *B. uniformis* and the amount of α-muricholic acid (α-MCA), taurolithocholic acid (TLCA), and UDCA in feces (Fig. 5c, d). A nearby, but not overlapping, locus on Chr 7 (gMxB4 QTL) was associated with *Bacteroides* and two fecal secondary BAs: 7-keto lithocholic acid (7-keto-LCA) and 7-keto deoxycholic acid (7-keto-DCA)/all ratio (Fig. 5c, d).

Different studies have demonstrated the roles of gMxB in the development of non-communicable diseases[7,22,46], such as metabolic dysfunction-associated fatty liver disease[47] or colorectal cancer[48]. To dissect the effects of the identified gMxB on metabolic traits, we first checked the correlation of *Turicibacter, Bacteroides*, as well as their species *T. sanguinis* and *B. uniformis*, with phenotypic traits collected in the BXD postprandial cohort. The abundance of *Turicibacter* and *T. sanguinis* was positively correlated with cecum weight (both absolute and relative to body weight (BW)) and BW loss, measured as the difference in BW after an overday fasting at 20 weeks of age (Fig. 5e). Conversely, they were negatively correlated with subcutaneous (ScWAT) and epididymal white adipose tissue (eWAT) mass under CD (Fig. 5e). In contrast, *Bacteroides* and *B. uniformis* abundances were negatively correlated with cecum weight as well as to BW loss under CD. Additionally, *Bacteroides* and *B. uniformis* were negatively associated with spleen weight (relative to BW) and *B. uniformis* showed a positive association with eWAT in HFD (Fig. 5e). These findings are in line with our previous observation that the *Turicibacter* abundance was higher in CD-fed (lean) mice, whereas *Bacteroides* was more abundant in HFD-fed (obese) mice (Fig. 2a, b). In addition, the correlations between bacterial abundance and the indicated metabolic traits were diet-specific, with HFD eliminating most of these associations. This further highlights the significant impact of diet on gut microbiome composition and its potential effect on host metabolism.

## Prioritizing the potential regulators of gMxB and unveiling their biological roles in humans

To investigate the candidate genes that potentially regulate these gMxB, we first filtered genes under these four gMxB QTLs with known segregating variants in the BXD strains predicted to have moderate/high impact. Then, genes enriched or expressed in the intestine obtained from the human protein atlas[49] (https://www.proteinatlas.org/humanproteome/tissue/intestine) were extracted. For the four identified gMxB QTLs, we identified 9, 6, 48, and 1 gene(s), respectively (Fig. 6a, b). To further prioritize these candidate genes, we retrieved suggestive ($p < 10^{-4}$) and significant ($p < 10^{-8}$) genetic variant-bacterium associations of human GWAS results from MiBioGen[50]. We then performed Mendelian randomization (MR) analysis to explore the causal effects of the gene expression of candidates in the human colon on the corresponding bacteria (BH-adjusted $P < 0.05$) using summary statistics from GTEx[51] and MiBioGen[50] (Fig. 6a, b). For genes under gMxB1 QTL, MR suggested that the gene expression of prostaglandin reductase 1 (*PTGR1*) in human sigmoid colon increases fecal *Turicibacter* abundance (Beta = 0.09 and BH-adjusted $P = 0.039$, Fig. 6b) with no significant heterogeneity observed (Cochran's Q test $p = 0.8$) in the sensitivity analysis. For genes within gMxB2 QTL, we found that genetic variants within protein tyrosine phosphatase receptor type D (*PTPRD*) are suggestively associated with the abundance of *Turicibacter* in human GWAS analysis (Fig. 6b). However, *PTPRD* did not have a significant *cis*-eQTL in human colons, preventing us from exploring the causality of the gene expression of *PTPRD* (Fig. 6b). In addition to analyzing these genes in the colon (site of primary BA transformation), we also examined their expression in liver (site of

primary BA production) datasets[52]. Notably, the liver expression of these two candidate genes was positively correlated with the genes involved in BA metabolism (Supplementary Fig. 4d, e). Overall, the associations between *Turicibacter* and these two genes, along with their predicted roles in BA metabolism, highlight *Ptgr1* and *Ptprd* as candidate mediators of the effects of gMxB1 and gMxB2, respectively, under CD (Fig. 6b). Under HFD condition, 19 candidate genes with *cis*-eQTLs in human colons were identified under the gMxB3 QTL. However, we could not further shortlist them as they did not associate with *Bacteroides* or *B. uniformis* in human GWAS and MR analyses. For the gMxB4 QTL, only one gene, gamma-aminobutyric acid type A (GABA$_A$) receptor subunit beta3 (*Gabrb3*), met 3 of our selection criteria (Fig. 6b) and could be a potential modulator regulating gMxB4, despite lacking human validation.

To better understand the roles of the three putative BA-microbiome co-regulators, *PTGR1, PTPRD*, and *GABRB3* in humans, we examined their associations with clinical traits and diseases in three populations—the FinnGen study[53], the Million Veteran Program (MVP)[54], and the UK Biobank (UKBB)[55,56]. While variants within *PTGR1* were suggestively related to colorectal cancer in the FinnGen study (Fig. 6c), genetic variants within *PTPRD* and *GABRB3* genes were associated with several metabolic disorders, such as diabetes, or liver fibrosis, in the FinnGen and MVP studies (Fig. 6c, d). Moreover, genetic variants within these three candidates were suggestively ($p < 10^{-4}$) associated with several metabolic traits in the UKBB, including cholesterol-related traits and fat mass (Fig. 6e–g and Supplementary Fig. 4a–c). Genetic variants within *PTPRD* were suggestively associated with heart failure in all three human populations, an effect that may be mediated by *T. sanguinis* abundance, as well as CA, CDCA, or TCA levels. Of note, *T. sanguinis* strains have previously been linked to cardiovascular disease (CVD) risk factors, such as serum cholesterol level[57]. Similarly, administration of CDCA or CA has been shown to reduce plasma triglycerides[58], which may explain the effect of the gMxB on CVDs. Overall, these data, together with the co-localized QTL results, suggest that alteration in the BA-gut microbiome crosstalk can potentially influence metabolic or cardiovascular health.

## Discussion
BA-microbiome crosstalk is known to play a crucial role in maintaining metabolic homeostasis[6,10,11], but the effects of diet, genetics, and their interaction on the gMxB are not fully elucidated. To gain more insight into the gMxB, GWAS have been previously applied to analyze the co-mapping genetic variants associated with both gut microbiome and plasma metabolites, including BAs, in a Japanese population. However, no genetic variants with pleiotropic associations were found[59]. This may be due to the presence of multiple confounding environmental factors in human populations, combined with the use of non-targeted metabolomics for measuring metabolite profiles. In contrast, the use of targeted BA analyses and controlled environmental factors (such as diet and living temperature) in a diversity outbred (DO) mouse population, allowed the identification of *Slc10a2*, a sodium/BA cotransporter, as a potential regulator of *T. sanguinis*-plasma CA interaction under high-fat high-sucrose diet[60]. However, this study did not investigate the effects of dietary factors and BA profiles in the liver, where primary BA are synthesized. Building on existing knowledge, we utilized a different mouse genetic reference population to evaluate cecal microbiome-BA interactions across two diets and three distinct biological compartments. We captured BA dynamics in enterohepatic circulation and systemic spill-over[6,61] in the postprandial state, as feeding serves as the primary trigger for BA release in the intestine and subsequent enterohepatic BA cycling. To better identify genetic, dietary, and GxE effects on the gMxB, we profiled the cecal microbiome and colon transcriptome, and integrated previously analyzed BA data in the plasma, liver, and feces[9] in postprandial BXD mice.

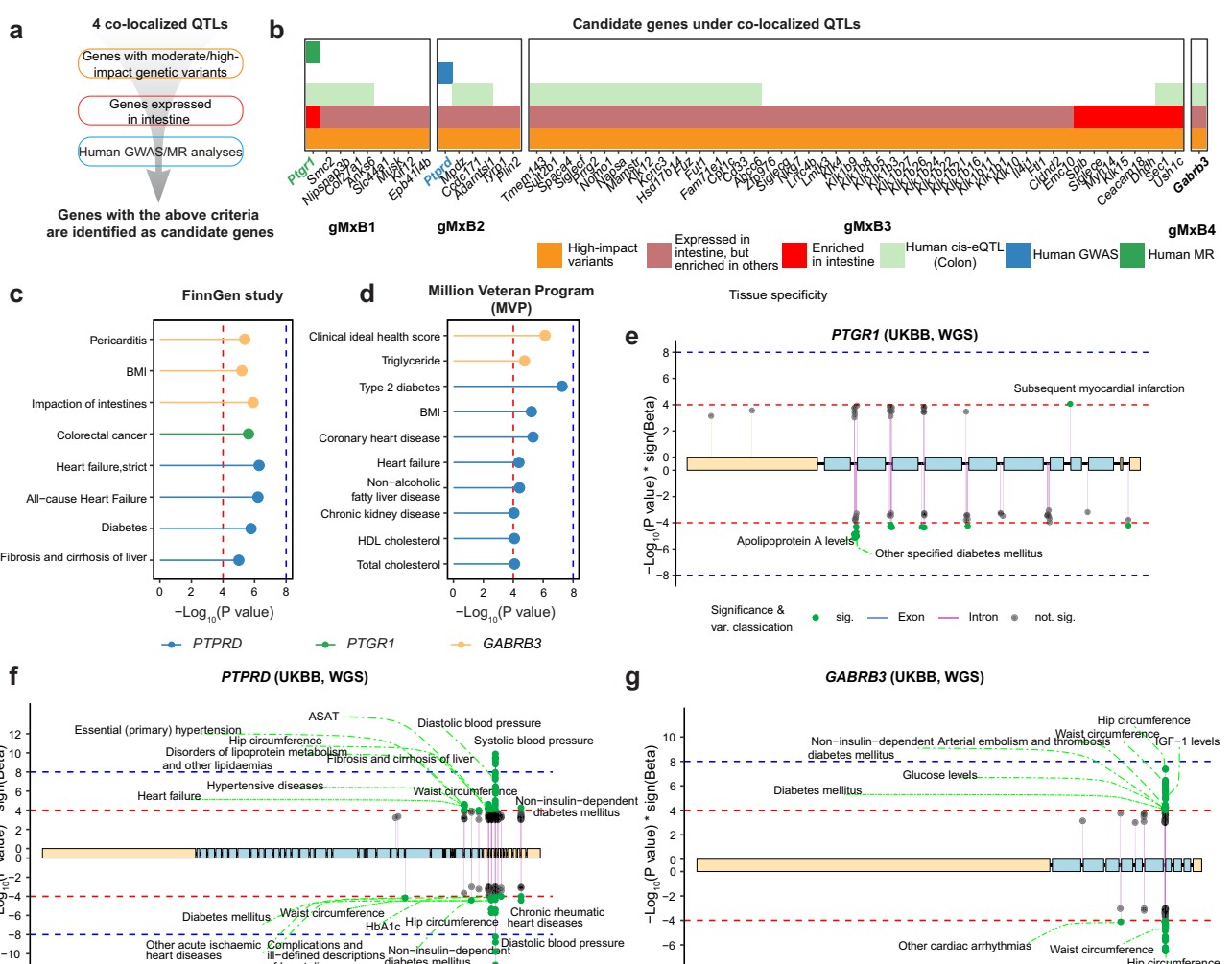

**Fig. 6 | Prioritization of candidate genes associated with gMxB and exploration of their potential roles in metabolic health. a** The filtering criteria for selecting candidate genes under the four co-mapping QTL peaks of gMxB. **b** The identified candidate genes under the four co-mapping QTL peaks of gMxB. Dot plots showing the metabolic health-related clinical traits that have GWAS hits within/around the three candidate genes in the FinnGen study[53] (**c**) and the Million Veteran Program (MVP)[54] (**d**). BMI: Body Mass Index. Lollipop plots showing the associations between metabolic traits and genetic variants in *PTGR1* (**e**), *PTPRD* (**f**), and *GABRB3* (**g**) obtained from GWAS results in the European population of the UKBB based on the whole genome sequence (WGS) data[90]. Only suggestive (-Log$_{10}$(P value) > 4, red dashed line) or significant (-Log$_{10}$(P value) > 8, blue dashed line) genetic variant-phenotype associations are highlighted in green, and the most significant associations were labeled. ASAT: Aspartate aminotransferase. Source data are provided as a Source Data file.

In line with previous studies[8,9], we found that genetic and dietary factors shape the gut microbiome[8] and BA composition[9], with genetic background also influencing the response to HFD challenge[8,9]. Most microbiome-BA interactions identified in this study were diet-dependent, highlighting the profound regulatory effects of dietary changes on gut homeostasis. At the transcriptome level, an interaction between bacterial species and inflammation-related genes was only observed in HFD (C3_HFD and C7_HFD CCA components). Notably, two bacterial species that we identified in C3_HFD or C7_HFD, *Bacteroides acidifaciens*[62] and *Lactobacillus taiwanensis*[63], are known to be associated with gut inflammation-related diseases[62,63]. Together, these findings suggest that the changes in the interaction between the gut microbiome and the host transcriptome upon HFD may arise because of intestinal inflammation induced by an HFD challenge, as previously reported[36,64].

To better understand the genetic and dietary influences on microbiome-BA interactions, we performed QTL mapping of both cecal microbiome and BA profiles, revealing four co-mapping genetic loci associated with gMxB in BXD mice. Under CD, we found a genetic locus associated with both the abundance of *T. sanguinis* and the

plasma CA levels that is in agreement with a previous DO mouse study[60]. In addition, the design of our study allowed us to also link this bacterium to plasma CDCA and TCA levels, as well liver CA levels. Certain *T. sanguinis* strains (such as MOL361, 18F6, and GALT-E2) are known to deconjugate TCA and glycochenodeoxycholic acid (GCDCA) to CA and CDCA, respectively[57,60]. The deconjugation role of *T. sanguinis* explained the positive correlation between the abundance of *T. sanguinis* and the relative amount of CA and CDCA under CD in our study. However, the function of *T. sanguinis* in transformation from GCDCA to CDCA is most likely less relevant in mice due to the low level of GCDCA that could only be barely detected in the feces in our data. In addition, two gMxB QTLs located on Chr 7 were identified under HFD. One QTL peak was associated with the abundance of *Bacteroides* and the amount of two fecal secondary BAs: 7-keto-LCA and 7-keto-DCA/All. Another nearby QTL peak was associated with its species, *B. uniformis*, as well as α-MCA, TLCA, and UDCA levels in feces. Of relevance, *B. uniformis* significantly influenced α-MCA, UDCA, and 7-keto-LCA colonic levels in a dextran sulfate sodium salt (DSS)-induced colitis mouse model[65], a model of chemically-induced intestinal inflammation that partially shares a transcriptomic signature with HFD-induced

intestinal changes[36]. Another isolated strain of *Bacteroides, Bacteroides intestinalis* AM-1, has also been reported to convert CA and CDCA into 7-keto-DCA and 7-keto-LCA, respectively, through its 7α-hydroxysteroid dehydrogenase (7α-HSDH) activity[66]. This evidence illustrates the reliability of our systems genetics approaches in identifying the communication between bacteria and BAs.

Furthermore, these identified gMxB have significant implications for host metabolic homeostasis. In particular, the gMxB related to *Turicibacter/T. sanguinis* are of interest, as CA and CDCA supplementation has been shown to reduce HFD-induced obesity in mice[67,68]. *Turicibacter* has previously been found to be negatively correlated with body weight in mice[69] and serum triglyceride level was reduced in *T. sanguinis*-monocolonized mice[70]. Consistent with these findings, our study links the abundance of these bacteria to lower adipose tissue mass, further supporting their potential role in metabolic regulation. Moreover, the identified potential mediators of the interaction between *Turicibacter/T. sanguinis* and these BAs, *Ptgr1* and *Ptprd*, can contribute to body composition changes. *Ptgr1* encodes the NADPH-dependent alkenal/one oxidoreductase and male *Ptgr1* knock-out mice display increased fat mass and decreased lean mass based on data collected by the international mouse phenotyping consortium (IMPC)[71] (https://www.mousephenotype.org/data/genes/MGI: 1914353). The associations between genetic variants within *PTGR1* and high-density lipoprotein (HDL) cholesterol and apolipoprotein A levels found in the UKBB further substantiate the potentially important role of *PTGR1* in metabolic control. In addition, *Ptprd* knock-out mice have decreased body weight and fat mass[72]. Genetic variants within *PTPRD* were reported to be associated with type 2 diabetes[73], which we also observed in the human FinnGen and MVP cohorts. Together, these results suggest that *PTGR1* and *PTPRD* could induce metabolic changes, and that they might do so by affecting the crosstalk between *T. sanguinis* and CA, CDCA, or TCA in the plasma or liver.

For the gMxB identified under HFD, *Gabrb3* was the most likely candidate under the QTL of *Bacteroides*, which encodes a GABA$_A$ receptor subunit, that potentially regulates *Bacteroides* abundance as well as 7-keto-DCA and 7-keto-LCA fecal levels. GABA$_A$ is a receptor of the major inhibitory neurotransmitter GABA and *Bacteroides* is known to regulate GABA production in the human intestine[74]. Variants within *GABRB3* influence the sensitivity of GABA$_A$[75] and dysfunctions in GABA signaling are involved in the development of neurological disorders, and can also affect immune responses and glucose metabolism[76]. Taken together, these observations from literature suggest that *Gabrb3* might potentially affect metabolic changes (and disease) through this bacterium-BA interaction. For the QTL peak of *B. uniformis*, this bacterium also regulates three non-12-OH BAs, α-MCA, TLCA, and UDCA and the proportion of non-12-OH BAs has been shown to be associated with reduced susceptibility to obesity[20,21]. This suggests that this gMXB QTL is also potentially involved in metabolic homeostasis by affecting the crosstalk between the three BAs and *B. uniformis*.

In conclusion, we leveraged the BXD mouse population to uncover the genetic, dietary, and GxE effects on the gMxB and dissected four gMxB QTLs. Among these, we identified three putative gMxB-modulating genes—*Ptgr1*, *Ptprd*, and *Gabrb3*. By integrating human datasets, we further confirmed *PTGR1* and *PTPRD* as potential regulators of gMxB and showed that they may affect metabolic health in humans. These results not only illustrate the effects of BA-gut microbiome interactions on metabolic homeostasis, but also provide insights about the role of host genetics and dietary factors on these interactions.

## Limitations of the study
In our study, we found that HFD challenge has significant effects on gut microbiome-BA interaction when compared to CD feeding. While the primary difference between the two diets is fat content, other factors, such as fiber content, may also contribute to the observed effects[77] and further play a role in shaping gMxB interactions. Additional research, beyond the scope of the current manuscript, could help clarify the individual contributions of these dietary factors. Our 16S rRNA sequencing approach successfully identified key microbial taxa and a subset of bacterial species that are involved in gMxB interaction. Complementing this approach with metagenomics in future research could provide even greater taxonomic and functional resolution. To correlate the 16S data with the dynamics of postprandial BA cycling and compartmentalization, we leveraged BA data from the same mice, where plasma and fecal BAs were collected at 20 and 24 weeks and liver BAs were profiled at the end of the study. These measurements provide valuable insights into BA metabolism over time. Expanding sampling to include portal blood and additional time points in future studies could further refine the understanding of real-time gut microbiome-BA interactions. Finally, despite the known differences in the composition of the BA pool between mice and humans, the majority of the BAs and bacterial QTLs found in this study is conserved across species, indicating that the uncovered gMxB using 32 BXD strains is sufficient to capture relevant genetic variation. However, incorporating female mice and/or mouse strains from different genetic reference populations as well as experimental validation in future studies could enhance the ability to mirror human genetic heterogeneity and uncover further genetic influences on gMxB, which further strengthens translational relevance to human populations. Despite these limitations, our experimental setup successfully highlighted robust gMxB interactions, partially supported by existing literature, and potentially relevant for the regulation of human metabolism.

## Methods
All animal experiments comply with all relevant ethical regulations and they were approved by the Swiss cantonal veterinary authorities of Vaud under the license 2257.2.

### Mouse experiments
The BXD strains were obtained from the University of Tennessee Health Science Center (Memphis, TN, USA) and have been maintained at the Center of PhenoGenomics (SV-CPG), École Polytechnique Fédérale de Lausanne (EPFL), for more than 20 generations. All mice were housed under 12 h light/dark cycle (lights on at 7am), with a temperature of 22 °C ± 1 °C with 30–50% humidity. Approximately 360 male mice from 34 BXD strains and 2 parental strains (C57BL/6 J and DBA/2 J) were included in the postprandial study. Mice were fed for 21 weeks (from 8 weeks old of age) either the Teklad Global 18% Protein Rodent Diet 2018 chow diet (CD – 24% kCal of protein, 18% kCal of fat, 58% kCal of carbohydrate – Envigo, Indianapolis, USA) or the Teklad Custom Diet TD.06414 high-fat diet (HFD – 18.3% kCal of protein, 60.3% kCal of fat, 21.4% kCal of carbohydrate – Envigo, Indianapolis, USA) from 8 weeks of age. Prior to euthanasia, mice were fasted overday and refed with their respective diet for 4 h following their physiological cycle[9]. Mice were anesthetized by 4.5–5% isoflurane followed by 1–1.5 ml blood collection form the heart. When the heart stops to beat, perfusion was then initiated by injection of 15–25 ml PBS via the left ventricle. When the liver turned pale/yellowish (sign of good perfusion), cervical dislocation was done. Multiple organs were then collected for future analysis. BXD strains used in this study: BXD1, BXD11, BXD12, BXD27, BXD32, BXD34, BXD39, BXD40, BXD43, BXD45, BXD48, BXD48a, BXD49, BXD51, BXD55, BXD6, BXD62, BXD64, BXD66, BXD67, BXD69, BXD73, BXD75, BXD79, BXD8, BXD81, BXD84, BXD87, BXD89, BXD90.

### Bacterial 16S rRNA gene amplicon sequencing
Mouse cecal content from 32 BXD strains described above was collected directly following euthanasia and stored at −80 °C. Cecal bacterial community composition was assessed by high-throughput

sequencing of the V1-V2 hyper-variable region of the bacterial 16S rRNA gene as previously described[78], following the basic protocol "Bacterial 16S rRNA sequencing for bacterial communities present in intestinal contents of mice" with PCR primers 27 F/338 R. 4−5 samples per strain and per condition were processed in 4 sequencing runs on an Illumina MiSeq platform using Paired End (PE) v2 2×250 chemistry. FASTQ files were demultiplexed using the Illumina-utils python package (version 2.7)[79].

## Microbiome data analysis

Data analysis was performed using R (v4.2.0). Reads were processed into a table of amplicon sequence variants (ASV) using the dada2 pipeline (R dada2 package version 1.18)[80] from FASTQ files. Briefly, reads were filtered and trimmed with parameters truncLen = c(170, 230), maxN = 0, maxEE = c(3, 2), truncQ = 2. A sequencing error model was generated, sequences were dereplicated, ASVs were inferred, paired-end reads were merged, and chimeric sequences were removed. ASVs were assigned a taxonomy at the species level using the Ribosomal Database Project (RDP) training set (release 19, https://doi.org/10.5281/zenodo.14168771)[81] based on exact sequence matching with parameter minBoot = 45. Samples with less than 6,000 sequencing reads (2 samples) were discarded. Alpha diversity of each sample was obtained through vegan R package (version 2.6-4). ASV counts were then normalized using the Phyloseq R package (version 1.38.0)[82]. Further, principal coordinate analysis (PCoA) and permutational multivariate analysis of variance (PERMANOVA) were performed based on Bray-Curtis dissimilarity using vegan R package. Differential abundance analysis was conducted by ANCOMBC R package (version 2.4.0)[83].

## Transcript profiling and analyses

We collected proximal colons for transcriptome analysis as previously described[36]. Briefly, we pooled ~5 samples with equal mass from the same strain and the same diet for RNA extraction. Total RNA was extracted using Directzol (Zymo Research) including the DNase digestion. RNA was amplified using the Ambion® WT Expression Kit (Life Technologies). cDNA was fragmented and labeled using the Affymetrix WT terminal labeling kit following manufacturers protocols. Microarrays were run using the Affymetrix mouse Clariom S Assay (GPL23038). apt-probeset-summarize from the Array Power Tool (APT) suite (v2.11.3) with the gc-sst-rma-sketch standard method were used to preprocess microarray data and resulting expression values were log-transformed. We ignored microarray probes targeting polymorphic regions in the BXD population. For probesets targeting the same transcript, the median value of the probes was considered. Principle component analysis (PCA) was performed by FactoMineR package (version 2.9)[84], and differential expression analysis was calculated using limma R package (version 3.50.3)[85].

## Procrustes analysis

To evaluate the overall relationship between host colon gene regulation and cecal microbiome composition under CD or HFD, we conducted a Procrustes analysis using vegan R package (version 2.6-4). For each condition, we applied Aitchison's distance to colon transcriptome data and Bray-Curtis distance to gut microbiome data as inputs for the Procrustes analysis[86].

## Multi-omics datasets integration

We applied sparse Canonical Correlation Analysis (CCA)[41,42], an established method to integrate gut microbiome and colon transcriptome and obtain the associations between the host gene group and bacteria genera group following the tutorial indicated in the Blekhman lab GitHub (https://github.com/blekhmanlab/host_gene_microbiome_interactions)[40]. For each diet, only bacteria identified to at least the genus level and bacterial species present in at least 10

mouse strains were included. Then, we performed sparse CCA separately using the PMA R package (version 1.2-2). Over-representation analysis (ORA) was further performed using cluster-Profiler R package (version 4.2.2)[87] with parameters minGSSize = 10 and maxGSSize = 1500. The Hallmark and Reactome genesets used in the ORA were retrieved through the msigdbr R package (version 7.5.1)[88].

The BA profiles in the plasma, feces, and liver were downloaded from Li et al.[9] (https://mpdpreview.jax.org/projects/Schoonjans1). Lasso penalized regression was performed separately in each diet using bacterial species as predictors and BAs as responses. This analysis was performed using the glmnet R package (version 4.1-8) following the tutorial also indicated in the same GitHub (https://github.com/blekhmanlab/host_gene_microbiome_interactions)[40].

## QTL mapping analyses on gut microbiome and BAs

The QTL mapping was performed with R qtl2 package (version 0.36)[89] based on the latest version of the BXD genotypes from GeneNetwork (http://files.genenetwork.org/current/GN600/BXD_current_rev050423.geno). Kinship between the 32 BXD strains was evaluated using the leave-one-chromosome-out (LOCO) method. Gut microbiome and BAs were normalized by quantile method (with non-positive values) or Box-Cox transformation (with only positive values). A genome scan was then performed using the scan1 function within each diet and the threshold ($p < 0.05$) of each QTL mapping was estimated based on permutation tests using the scan1perm function with 10,000 repeats. The significant QTL peaks were calculated by the find_peaks function with parameters: prob = 0.95 and peakdrop = 0.5. Colocalized QTL peaks are identified if the 95% confidence intervals of these QTLs were overlapped.

## Human dataset analysis

We assessed to metabolic-related phenotypic traits from the UKBB[55,56] under Application Number 48020 and European-ancestry participants (based on the UKBB return dataset 2442) were extracted for further GWAS analysis using REGENIE. Step1 involves the estimation of population structure using genotyping arrays provided by UKBB. Step2 calculated genetic variant-phenotype associations based on the whole exome sequencing (WES) data, using the following covariates: the first 10 genetic principal components, age, sex, age:sex interaction, and Body Mass Index (BMI). The GWAS summary statistic calculated based on UKBB whole genome sequencing (WGS)[90] were downloaded from GWAS Catalog[91]. The summary statistic from the FinnGen[53] study were downloaded from https://www.finngen.fi/en/access_results, and those for MVP[54] database were accessed through dbGaP under accession phs002453.v1.p1 through AgingX project (ID: 10143). The significant ($p < 10^{-8}$) or suggestive ($p < 10^{-4}$) associated phenotypic traits of genetic variants within candidate genes in each human population were further extracted.

## Mendelian randomization analysis

Summary statistics-based Mendelian randomization was used to estimate the causal effects of the tissue-specific expression of candidate genes on various outcomes. Significant *cis*-eQTLs of candidate genes in the sigmoid colon and transverse colon were selected and their effect sizes were obtained from the GTEx Portal[51] on 2023-03-28 (v8, https://www.gtexportal.org/home/datasets). GWAS summary statistics for phenotypes of interest, namely the bacterial genera *Bacteroides* and *Turicibacter* (based on MiBioGen database) and their species *B. uniformis* and *T. sanguinis*, were obtained from the IEU OpenGWAS project[50,92], using the ieugwasr package (version 0.1.5). In most cases, the summary statistics of multiple significant *cis*-eQTLs were also available in the outcome GWAS, enabling their use as instrumental variables (IVs). Because only *cis*-eQTLs were selected, the IVs present a high level of linkage disequilibrium (LD) between each other and LD

pruning would leave very few independent IVs, which would make the effect estimates less robust and liable to change based on which IVs are selected. Instead, we used an existing PCA-based approach[93] which combines IVs using LD-based PCA, yielding PC IVs which are independent by construction. The LD structure was estimated based on the genotype files from GTEx and extracted using PLINK v1.90b6.21[94]. We selected top PC IVs to explain 99% of the genetic variance and used the inverse-variance weighted approach[93] to estimate the causal effect. For results where the causal effect estimate was significant, we also performed a sensitivity test with a leave-one-out approach, removing one PC and estimating the causal effects using the remaining PCs. The resulting estimated causal effects were tested for heterogeneity using Cochran's Q test. The list of IVs is also provided (Supplementary Data 1). In cases where only a single IV was available in both GTEx and the outcome GWAS (or if a single PC was retained), we used the Wald ratio method in TwoSampleMR[95].

Because the effect sizes from GTEx[51] do not have a direct interpretation or unit, the magnitude of causal effect estimates are not meaningful, however the directionality of the effect remains informative. The scale of these QTL effects remains constant per gene, however, which enables their combination using the above-mentioned PCA-based approach[93].

## Reporting summary

Further information on research design is available in the Nature Portfolio Reporting Summary linked to this article.

## Data availability

The colon transcriptome data generated in this study have been deposited in the GEO database under accession code: GSE272489. The cecal 16S ribosomal RNA sequencing generated in this study have been deposited in the NCBI Sequence Read Archive (SRA) under accession code PRJNA1137099 (https://www.ncbi.nlm.nih.gov/sra?linkname=bioproject_sra_all&from_uid=1137099). The phenotype and bile acid data used in this study are available in the Mouse Phenome Database: https://phenome.jax.org/projects/Schoonjans1. All data supporting the findings described in this manuscript are available in the article and in the Supplementary Information and from the corresponding author upon request. Source data are provided with this paper.

## Code availability

All analyses were performed using R version 4.4.0, employing the tools and packages described in the Methods section. No original code was developed for this study; all software and packages used are publicly available through Bioconductor and/or GitHub.

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

## Acknowledgements

We thank the Auwerx and Schoonjans lab members for technical assistance and discussions. We acknowledge the help of Maroun Bou Sleiman, Alexis Bachmann, Lorne A. Rose and Jesse F. Ingels for transcript profiling. The work was supported by the École Polytechnique Fédérale de Lausanne (EPFL to K.S. and J.A.) and grants from the Kristian Gerhard Jebsen Foundation (to K.S and J.A.), the European Research Council (ERC-AdG-787702 to J.A.), the Swiss National Science Foundation (310030-189178 and CRSII5-180317 to K.S. and 31003A-179435 and 310030-214826 to J.A.), and the Global Research Laboratory (GRL) National Research Foundation of Korea (NRF 2017K1A1A2013124 to J.A.). X.L. was supported by the China Scholarship Council (201906050019).

## Author contributions

The study was conceived by X.L., A.P., K.S and J.A.; H.L., A.P. and A.R. performed laboratory experiments. Data analyses were carried out by X.L.; J.S. and G.V.A. helped with the human genetic analysis. X.L., A.P. and J.A. wrote the original manuscript. J.D.M., Q.W., A.J., K.S and J.A. reviewed and edited the manuscript with contributions from all co-authors.

## Competing interests

The authors declare no competing interests.
