## [Transparent Peer Review file · Nature Communications]

Genetic and dietary determinants of gut microbiome-bile acid interactions in the BXD genetic reference population

Corresponding Author: Professor Johan Auwerx

Version 0:

Reviewer comments:

Reviewer #1

(Remarks to the Author)

Title: Genetic and dietary determinants of gut microbiome-bile acid interactions 1 in the BXD recombinant inbred mouse population

Journal: Nature communications

This study aimed to understand the interaction between gut microbiome and bile acids (gMxBs) influenced by genetics and diet. Using systems genetics and multi-omics data from 32 BXD mouse strains, the researchers identified four diet-dependent genetic loci and candidate genes, such as PTGR1, PTPRD, and GABRB3, potentially regulating gMxBs. The findings provide insights into gut microbiome-host communication and its impact on metabolic health.

1. Could the authors please explain the rationale for feeding the mice for a duration of 21 weeks? Is there a specific reference or study that justifies this choice? Additionally, why were plasma and fecal samples not collected in the final week of feeding? Given that bile acid profiles can vary significantly over time and across tissues, sampling at the end of the feeding period might provide a more accurate representation of the long-term effects of diet.

2. The authors have utilized bile acid profiles data from a previous study. Could more information be provided on how this data was integrated and validated within the current study's context? Is it reliable to assume that the bile acid profiles from a previous study will accurately reflect the conditions and outcomes of the current investigation?

3. The Figure 1E could not present the data that the abundance of Firmicutes was increased upon HFD in the C57BL/6J strain, while it was decreased in the DBA/2J. I suggest that the ratio of Firmicutes to Bacteroidetes must be presented in each column. Could the authors provide an explanation for the variable effects of high-fat diet on the Firmicutes population in the gut microbiome of mice with different genetic backgrounds?

4. It is suggested that the authors re-analyze the 16S rRNA amplicon metagenomics data at the operational taxonomic unit (OTU) level, rather than solely at the genus level, and attempt to annotate the OTU representative sequences to the species level where possible.

5. This study extensively utilized bioinformatics and computational analyses, integrating multi-omics data such as microbiome profiles, transcriptomics, and bile acid levels from the BXD mouse strains. I would request the authors to provide experimental validation for the bioinformatics findings, such as in vitro or in vivo assays to confirm the role of identified candidate genes in gut microbiome-bile acid interactions.

Reviewer #3

(Remarks to the Author)

In the manuscript entitled "Genetic and dietary determinants of gut microbiome-bile acid interactions in the BXD recombinant inbred mouse population" Xiaoxu Li and colleagues describe data on gut microbiome-bile acid crosstalk in 32 postprandial BXD mouse strains. Three genes were highlighted as potentially involved in gut microbiome-bile acid crosstalk. Finally, the potential relevance of these genes for human metabolic traits was assessed using the UK biobank, FinnGen and million veteran population cohorts. Sound methodology was used and the conclusions are generally well-supported by the data.

Altogether, this comprehensive study illustrates the potential impact of the host genetics and dietary factors on the gut microbiome-bile acid interactions that can affect metabolic health and therefore considerably advances research in this rapidly developing field. There are, however, several issues that warrant attention and/or clarification.

- For the current study, 32 BXD mouse strains were used, while many more have originally been generated and also more were used in a recent study reference 9 (Li H, et al. *Cell Metab*, 2022). Although it is understandable that only a subset of BXD strains were used for the experiments described in the current manuscript, it should be outlined based on what criteria the strains that were used have been selected.

- In the experiments, chow diet was compared to a high-fat diet. However, the latter diet does not only contain high amounts of fat, but also has a low fiber content which is solely composed of cellulose. Hence a regular chow diet does not represent a proper control diet to study the effects of a high fat content of the diet when the latter diet is a synthetic diet (Pellizzon MA, Ricci MR, *Nutr Metab (Lond)*, 2018). The different fiber content and composition of the diets may actually have a greater impact on the gut microbiome composition than the fat content. Also, the lack of fiber rather than the high fat content in the applied HFD is likely to underly the reduced size of the cecum of HFD-fed mice.

- What were the criteria used to label genes as 'highly expressed' in the intestine for filtering genes that potentially regulate gMxBs (as shown in Fig 6b)? On page 9, line 297 and 298, it is stated that GABRB3 is highly expressed in human intestines. According to <https://www.proteinatlas.org>, expression of GABRB3 is about 1 nTPM in human intestines, which is considerably lower as compared to many other tissues, including gallbladder (13.2 nTPM) and cerebral cortex (45.7 nTPM).

- Correlations between bacteria and bile acids in plasma, liver and feces are provided. Given the high first-pass extraction of bile acids by the liver as well as the activity of CYP2A12 (which rehydroxylates secondary bile acids and thereby converts them back into primary species) in mouse livers, plasma obtained from portal blood would be ideal to assess gut microbiome-bile acid interactions. If this is not available, bile acids measured in cecum contents might represent the second best option and be more informative than liver, feces and peripheral plasma.

- Were the correlations in Fig. 4C and D between bile acids and bacterial abundance made using absolute bile acid concentrations or bile acid composition (% of total bile acids) in the different compartments? The latter would be preferred.

- On page 11, lines 351-355, it is stated that "Certain strains of *Turicibacter* (such as MOL361, 18F6, and GALT-E2) can deconjugate TCA to CA and glycochenodeoxycholic acid (GCDCA) to CDCA, which explains the positive correlations observed between the abundance of *Turicibacter* and the amount of CA and CDCA under CD condition in our study". However, GCDCA levels are generally extremely low in mice. Hence, it is not very likely that deconjugation of GCDCA represents a quantitatively meaningful source of CDCA.

- The correlations between gut bacteria and a limited amount of metabolic traits are shown in Fig 5F. Did the authors correlate gut bacteria with more metabolic traits (e.g. plasma lipids, transaminases etc.)? If so, the authors are strongly encouraged to provide correlations between the selected gut bacteria and all metabolic traits tested in the data supplement. Also, spleen, soleus and cecum weights are expressed as percent of BW, making the latter a potential confounder since the correlations with BW are not provided (only with BW loss during BA kinetics).

- The authors are encouraged to add a paragraph to the discussion section in which they describe the limitations of their study, including use of a limited amount of BXD strains, use of only a single sex (bile acid metabolism is different between male and female mice), differences between mice and humans and its potential impact with regards to translation of the results, etc.

Minor:

- Could authors explain the details regarding BW loss (BA kinetics), Fig. 5F? During what time period did this weight loss occur? It is rather cumbersome for future readers to have to read previous papers for these details.

- Please provide some detail concerning the impact of soleus weight as such on metabolism.

- Could the authors explain why significant eQTLs of candidate genes were selected in the sigmoid colon and transverse colon for MR analysis, while, in the mouse experiments the proximal colon was collected and used for transcriptome analysis.

- Page 11, lines 337-338: "In line with previous studies, we found that the genetic and dietary factors shape gut microbiome and BA composition and the response to HFD challenge." In short: "...dietary factors shape... response to HFD challenge". High-fat already represents a dietary factor. Hence, this statement does not seem to be appropriately formulated. Or are the authors referring to dietary factors other than the high-fat content in this sentence?

Reviewer #4

(Remarks to the Author)

Version 1:

Reviewer comments:

Reviewer #3

(Remarks to the Author)

All my concerns have been addressed in a satisfactory manner and I have no further questions.

Reviewer #4

(Remarks to the Author)

Reviewer #5

(Remarks to the Author)

The authors have clearly improved the quality of the manuscript during the revision process. This reviewer is satisfied with the authors' responses and the new data provided.

Reviewer #6

(Remarks to the Author)

In this work the authors seek to characterise the effects of high-fat diet on the mouse microbiome, trying to understand if any genetic variants and genes are involved in influencing these effects. The authors then try to use human genetic datasets to validate their results and link them to human health.

As my expertise is mostly on human genetics I will focus my comments on this section and leave to the other reviewers the others.

Overall the human genetics part of the paper may make sense in design where the authors try to reproduce their mice results in human verifying if differences in expression of the identified genes in the colon may lead to differences in the microbiome, using publicly available summary data.

There are several issue with the whole procedure:

1) The MR analysis is not described at all. The only mention is that they used a PCA based method and a reference. No broad description of the method is present or why this was chosen amongst the dozen existing. The choice of genetic instruments is not described (which SNPs where used? how where selected? What was the power of the selected instruments? etc. etc). These information are fundamental to evaluate the results. Furthermore no sensitivity analysis was conducted to verify the sturdiness of the results. So although I think the idea is potentially good there needs the analysis needs to be conducted and reported in much more thoroughly and with many more details. This could be complemented with colocalization analysis which is quite standard now a days. This could be applied also to the disease outcomes to show potential mediation of the microbiome on disease.

2) Similarly the choices for the look up in GWAS is not clear as they looked specifically only to WES where imputed data exists and also WGS. Why did the authros focus only on coding variants given that their results showed an effect in differences of expression which is likely due to non-coding variants? Generally coding variants are analysed through aggregation tests (ie Burden/SKAT) as those with high impact will have very low frequency. For UKB there is even a public database with previously reported analses in UKB (GeneBass) which the authors could look for evidence of their results.

So overall I think that the aproach and results are interesting but that the analysis is still not deep and sturdy enough to answer to their scientific qesiton.

Version 2:

Reviewer comments:

Reviewer #6

(Remarks to the Author)

The authors have replied to my comments and I have not further ones.

General response to all reviewers:

We thank the reviewers for providing valuable comments and suggestions. We agree with Reviewer #1 that species-level annotation of operational taxonomic units (OTUs) could provide more precise insights into gut microbiome composition and its interactions with bile acids (BAs). However, as our microbiome data is based on 16S rRNA sequencing - offering lower taxonomic resolution than metagenomics - species-level annotation is inherently limited. In our original manuscript, we analyzed and annotated the 16S rRNA dataset using the SILVA reference database (version 138.1), the most widely used for 16S analyses. However, only 6.7% of OTUs could be classified at the species level (Figure R1), which is why our initial analysis was conducted at the genus level.

During revision, we reassessed the 16S dataset using both the updated SILVA and ribosomal database project's (RDP) reference databases (Figure R1). We found that the RDP reference provided species-level annotation for over 47% of the OTUs (Figure R1), allowing us to gain more insights at the species level compared to our original analysis. Consequently, we re-analyzed our 16S rRNA data using the RDP database. Importantly, this re-analysis provided additional insights while preserving the main conclusions of our study, as described below.

Figure R1: Bar plot showing the percentage of OTU annotation with the indicated databases at the genus (red) and species (blue) levels.

Of note, at the genus-level, results remained highly consistent between SILVA (version 138.1, in the original manuscript) and RDP (in the revised manuscript) analyses. For example, in the differential abundance analysis, *Lactococcus* remained the most increased genus upon high fat diet (HFD) feeding, while *Turicibacter* was significantly enriched in chow diet (CD) and barely detected in HFD (original Figure 2A and revised Figure 2a). Similarly, in a strain-dependent view, results also remained concordant: e.g., *Muribaculum* abundance was significantly increased only in BXD6 and BXD87 mice in both analyses (original Figure 2B and revised Figure 2b).

Original Figure 2:

Revised Figure 2:

Figure 2: HFD feeding alters the gut microbiome at the genus level. (a) Volcano plot showing the effect of HFD on the abundance of each bacterial genus across all BXD mouse strains. Genera with $\text{Log}_e(\text{Fold change}) > 1$ and q value < 0.05 , obtained from analysis of composition of microbiomes – ANCOM, were identified as differentially abundant genera (DAGs) and colored by the corresponding phylum. (b) Heatmap showing a subset of genera that were significantly changed (q value < 0.05 , ANCOM) upon HFD in both diet-independent view and in more than 15 BXD strains. q values are indicated as follows: * q value < 0.05 ; ** q value < 0.01 ; *** q value < 0.001 . Missing values in both diets were indicated by grey color. (c, d) Procrustes analyses indicating the overall association between variation in host colon gene expression and gut microbiome composition under CD (c) and HFD (d) based on the Aitchison's distance of host gene expression data (purple color, response) and Bray-Curtis distance of gut microbiome data (orange color, predictor). P: P value were calculated in Procrustes analyses.

In the QTL analysis, despite using the RDP database, we identified the same QTL peaks for *Turicibacter* and *Bacteroides* on chromosomes 4 and 7 under CD and HFD conditions, respectively. However, some minor differences emerged between the two databases. For

example, *Lachnoclostridium* is not classified as a genus in RDP, preventing exact replication of certain QTL results observed with SILVA (original Fig. S3 and revised Fig. S3). Nonetheless, our findings remain robust and consistent at the genus level, regardless of the database used. This consistency encouraged us to transition to the RDP reference database, which enhances species-level annotation.

Please find all revised figures and text (colored in red) in the revised manuscript

Original Figure S3:

Revised Figure S3 for QTL mapping of gut microbiome at the genus level:

Figure S3: QTL mapping analyses of gut bacterial genera and species in the BXDs. (a, b) Manhattan plots showing the associations between genetic loci and bacterial genera under CD (a) and HFD (b).

Reviewer #1

This study aimed to understand the interaction between gut microbiome and bile acids (gMxBs) influenced by genetics and diet. Using systems genetics and multi-omics data from 32 BXD mouse strains, the researchers identified four diet-dependent genetic loci and candidate genes, such as *PTGR1*, *PTPRD*, and *GABRB3*, potentially regulating gMxBs. The findings provide insights into gut microbiome-host communication and its impact on metabolic health.

1. Could the authors please explain the rationale for feeding the mice for a duration of 21 weeks? Is there a specific reference or study that justifies this choice? Additionally, why were plasma and fecal samples not collected in the final week of feeding? Given that bile acid profiles can

vary significantly over time and across tissues, sampling at the end of the feeding period might provide a more accurate representation of the long-term effects of diet.

The decision to feed the mice for 21 weeks was made to align our study with a related study in fasting BXD mice¹⁻⁵, enabling comparative analyses between the two datasets.

We agree that collecting plasma and fecal samples during the final week of feeding could have provided additional insights into the long-term effects of the diet. However, our study was specifically designed to capture the physiological postprandial elevation of bile acids in plasma and the 24-hour bile acid fecal pool. The strict Swiss animal legislation prohibits performing multiple phenotyping tests within the same week. To comply with these regulations, we staggered the tests throughout the study, making it unfeasible to perform all experiments in the final week. This information has now been included into the “**Limitation of study**” section (Page 13, Lines 394-417).

2.The authors have utilized bile acid profiles data from a previous study. Could more information be provided on how this data was integrated and validated within the current study's context? Is it reliable to assume that the bile acid profiles from a previous study will accurately reflect the conditions and outcomes of the current investigation?

We apologize for the confusion. The 16S microbiome data from the fed BXD mice are newly generated but originate from the same study and the same cohort of mice in which bile acids were previously profiled⁶. Therefore, these bile acid profiles presented in our study accurately reflect the conditions and outcomes of the current investigation. This clarification has now been incorporated in the revised text (Page 4, Lines 53-54).

3.The Figure 1E could not present the data that the abundance of Firmicutes was increased upon HFD in the C57BL/6J strain, while it was decreased in the DBA/2J. I suggest that the ratio of Firmicutes to Bacteroidetes must be presented in each column. Could the authors provide an explanation for the variable effects of high-fat diet on the Firmicutes population in the gut microbiome of mice with different genetic backgrounds?

Thank you for your suggestion. We have now calculated the ratio of Firmicutes to Bacteroidetes (F/B ratio) for all the BXD strains included in the study (Fig S1a). Our analysis shows that this ratio significantly increases in the C57BL/6J strain upon HFD, while it significantly decreases in the DBA/2J and remains unchanged in most other BXD mouse strains. The variable effects of HFD on this ratio across different mouse strains with distinct genetic backgrounds have also been reported previously⁷ and confirmed in humans⁸. These differences may stem from genetic influences on the gut microbiome, as well as strain-specific dietary responses. Please find the corresponding figure (Fig S1a) and statements in the revised manuscript (Page 5, Lines 87-91).

4. It is suggested that the authors re-analyze the 16S rRNA amplicon metagenomics data at the operational taxonomic unit (OTU) level, rather than solely at the genus level, and attempt to annotate the OTU representative sequences to the species level where possible.

Annotating operational taxonomic units (OTUs) at the species level could indeed provide more precise insights into gut microbiome composition and enhance our understanding of gut microbiome-bile acid interactions. We have re-analyzed the 16S rRNA dataset and now annotate the OTUs to the species level using the RDP reference database (see **General response to all reviewers**). The annotated bacterial genus and species were both used in this study, please find the updated figures and results in the revised manuscript.

5. This study extensively utilized bioinformatics and computational analyses, integrating multi-omics data such as microbiome profiles, transcriptomics, and bile acid levels from the BXD mouse strains. I would request the authors to provide experimental validation for the bioinformatics findings, such as in vitro or in vivo assays to confirm the role of identified candidate genes in gut microbiome-bile acid interactions.

We agree that experimental validation would be the most direct approach to confirm the role of candidate genes in gut microbiome-bile acid interactions. However, such validation would require extensive regulatory approval under Swiss animal legislation, a process that takes over four months. Moreover, to ensure robust conclusions, these experiments would need to be conducted across multiple genetic backgrounds and in mice fed either CD or HFD for 21 weeks, further increasing the complexity and duration of the study. For these reasons, we believe that these mechanistic validations fall beyond the scope of this first study, which primarily focuses on systems genetics analysis.

Nevertheless, we sought to assess the validity of our findings through existing literature. Notably, *Turicibacter sanguinis* has been shown to deconjugate TCA and GCDCA *in vitro*⁹, supporting our predicted functional links. Moreover, in a dextran sulfate sodium salt (DSS)-induced colitis mouse model, which shares transcriptome similarities with HFD-fed mice³, *Bacteroides uniformis* significantly influenced colonic levels of α -MCA, UDCA, and 7-Keto-LCA¹⁰. These independent findings provide additional support for our computational results and highlight the reliability of our systems genetics approach in identifying microbiome-bile acid interactions. We have now included these observations into the Discussion (Page 11-12, Lines 335-349) and acknowledge the limitation of experimental validation in the section of “**Limitation of study**” (Page 13, Lines 394-417).

Reviewer #3

In the manuscript entitled “Genetic and dietary determinants of gut microbiome-bile acid interactions in the BXD recombinant inbred mouse population” Xiaoxu Li and colleagues describe data on gut microbiome-bile acid crosstalk in 32 postprandial BXD mouse strains. Three genes were highlighted as potentially involved in gut microbiome-bile acid crosstalk.

Finally, the potential relevance of these genes for human metabolic traits was assessed using the UK biobank, FinnGen and million veteran population cohorts. Sound methodology was used and the conclusions are generally well-supported by the data. Altogether, this comprehensive study illustrates the potential impact of the host genetics and dietary factors on the gut microbiome-bile acid interactions that can affect metabolic health and therefore considerably advances research in this rapidly developing field. There are, however, several issues that warrant attention and/or clarification.

- For the current study, 32 BXD mouse strains were used, while many more have originally been generated and also more were used in a recent study reference 9 (Li H, et al. Cell Metab, 2022). Although it is understandable that only a subset of BXD strains were used for the experiments described in the current manuscript, it should be outlined based on what criteria the strains that were used have been selected.

The selection of BXD strains for the fed study⁶ was made to match those used in an extant study in fasted mice¹⁻⁵, ensuring consistency between the two datasets – see our response to the first question of Reviewer #1. While the fed study included 36 BXD strains (Li H, et al. Cell Metab, 2022), cecum samples were collected from only 32 strains, which aligns with the number of strains used in our current study. This is now specified in the Methods (Page 14, Lines 420-422 and Lines 431-432).

- In the experiments, chow diet was compared to a high-fat diet. However, the latter diet does not only contain high amounts of fat, but also has a low fiber content which is solely composed of cellulose. Hence a regular chow diet does not represent a proper control diet to study the effects of a high fat content of the diet when the latter diet is a synthetic diet (Pellizzon MA, Ricci MR, Nutr Metab (Lond), 2018). The different fiber content and composition of the diets may actually have a greater impact on the gut microbiome composition than the fat content. Also, the lack of fiber rather than the high fat content in the applied HFD is likely to underly the reduced size of the cecum of HFD-fed mice.

Thanks for your thoughtful comment. We selected these diets because they align with the diets used in a related BXD fasting study¹⁻⁵, allowing future comparative analyses between the two datasets. We agree that fiber is an important factor affecting gut microbiome, and it is indeed different in the chow diet (3.8% Crude Fiber and 14.7% Neutral Detergent Fiber) and high-fat diet (6.5% Cellulose) used in this project. We have added this in “**Limitation of study**” section (Page 13, Lines 394-417).

- What were the criteria used to label genes as ‘highly expressed’ in the intestine for filtering genes that potentially regulate gMxBs (as shown in Fig 6b)? On page 9, line 297 and 298, it is stated that GABRB3 is highly expressed in human intestines. According to <https://www.proteinatlas.org>, expression of GABRB3 is about 1 nTPM in human intestines, which is considerably lower as compared to many other tissues, including gallbladder (13.2 nTPM) and cerebral cortex (45.7 nTPM).

Thank you for pointing this out. We downloaded intestine-specific gene profiles from the human protein atlas and used genes that are elevated in the intestine or are elevated in other tissues but expressed in intestine to filter candidate genes under co-localized QTL peaks. We have corrected the annotation as “Expressed in intestine, but elevated in others” in revised Figure 6b and text (Page 9, lines 255-257) to avoid the confusion.

- Correlations between bacteria and bile acids in plasma, liver and feces are provided. Given the high first-pass extraction of bile acids by the liver as well as the activity of CYP2A12 (which rehydroxylates secondary bile acids and thereby converts them back into primary species) in mouse livers, plasma obtained from portal blood would be ideal to assess gut microbiome-bile acid interactions. If this is not available, bile acids measured in cecum contents might represent the second best option and be more informative than liver, feces and peripheral plasma.

We agreed that portal blood would have been ideal to measure BAs; however, we did not collect it (difficult procedure for more than 300 mice). Also, cecal BAs were not measured and we unfortunately could not profile cecal BAs for the revision of this manuscript due to the high number of additional samples and availability of the metabolomic facility. However, we are convinced that the fecal BAs are a good proxy for the cecal BA profile, as illustrated in the literature¹¹.

- Were the correlations in Fig. 4C and D between bile acids and bacterial abundance made using absolute bile acid concentrations or bile acid composition (% of total bile acids) in the different compartments? The latter would be preferred.

Thanks for pointing this out. We used absolute BA concentrations and the abundance of bacterial genera. We have now changed to use the relative abundance of each BA (expressed as a fraction of the total in each biological compartment: BA X /all BAs) and the abundance of bacterial species for the Spearman correlation and Lasso regression analyses. Please find the updated Figure 4c and 4d and result in the revised manuscript (Page 7-8, Lines 184-203).

- On page 11, lines 351-355, it is stated that “Certain strains of Turicibacter (such as MOL361, 18F6, and GALT-E2) can deconjugate TCA to CA and glycochenodeoxycholic acid (GCDCA) to CDCA, which explains the positive correlations observed between the abundance of Turicibacter and the amount of CA and CDCA under CD condition in our study”. However, GCDCA levels are generally extremely low in mice. Hence, it is not very likely that deconjugation of GCDCA represents a quantitatively meaningful source of CDCA.

We agree that GCDCA levels are generally extremely low in mice and we now have updated our statement. Please find the updated text in the revised manuscript (Page 11-12, Lines 337-342).

- The correlations between gut bacteria and a limited amount of metabolic traits are shown in

Fig 5F. Did the authors correlate gut bacteria with more metabolic traits (e.g. plasma lipids, transaminases etc.)? If so, the authors are strongly encouraged to provide correlations between the selected gut bacteria and all metabolic traits tested in the data supplement. Also, spleen, soleus and cecum weights are expressed as percent of BW, making the latter a potential confounder since the correlations with BW are not provided (only with BW loss during BA kinetics).

We did not measure plasma clinical chemistry in the current BXD fed study, preventing direct correlation of gut bacteria abundance with plasma lipids or transaminases (e.g., alanine aminotransferase (ALAT) and aspartate aminotransferase (ASAT)).

To explore potential associations between plasma clinical chemistry and the gut microbiome, we instead used plasma clinical chemistry data from a separate BXD study¹⁻⁵ conducted under the same diet duration but with tissue collection in a fasted state (see our response to the first question of Reviewer #1). We now performed Spearman correlation analysis between plasma parameters from the BXD fasting study¹⁻⁵ and the four bacteria identified in the current manuscript. However, no significant correlations were observed (Figure R2). Given that these two BXD studies were conducted at different times and under different feeding conditions (fed vs. fasted), potential biases may have influenced the correlation analysis, affecting the associations between plasma parameters and the gut microbiome. For these reasons, we decided not to include these data in the revised version of the manuscript.

Figure R2: Dot plot showing the absence of correlations between gut bacteria and plasma metabolic-related traits in the BXD panel, using data from BXD fasting study¹⁻⁵.

In addition, we correlated the bacteria with all possible traits in the current BXD study, including body weight, organ weight, and organ weight adjusted for body weight. The metabolic traits that significantly correlated with bacterial genera or species, such as *T. sanguinis*-cecum weight, were shown in Figure 5e in the revised manuscript. Please also find the revised text in **Page 9, Lines 236-246**. All results from correlation analyses, including non-significant results, were included in the source data file.

- The authors are encouraged to add a paragraph to the discussion section in which they describe the limitations of their study, including use of a limited amount of BXD strains, use of only a single sex (bile acid metabolism is different between male and female mice), differences between mice and humans and its potential impact with regards to translation of the results, etc.

We have now included all the points you and Reviewer #1 raised in the “**Limitation of study**” section that addressed these concerns (**Page 13, Lines 394-417**).

Minor:

- Could authors explain the details regarding BW loss (BA kinetics), Fig. 5F? During what time period did this weight loss occur? It is rather cumbersome for future readers to have to read previous papers for these details.

BW loss (BA kinetics) represents the difference in body weight between the start of the experiment and after over-day fasting. We have now clarified this in the revised manuscript (Page 9, Lines 240-241).

- Please provide some detail concerning the impact of soleus weight as such on metabolism.

We initially included soleus muscle weight because it is an oxidative muscle. However, we agree that it is a small muscle, prone to measurement errors. Therefore, we have removed it from the metabolic traits shown in the revised Figure 5e.

- Could the authors explain why significant eQTLs of candidate genes were selected in the sigmoid colon and transverse colon for MR analysis, while, in the mouse experiments the proximal colon was collected and used for transcriptome analysis.

We used eQTLs results from GTEx as “exposure” in the MR analysis. Since the GTEx database only provides data for two colon segments - sigmoid colon and transverse colon - we selected those for our MR analysis. While they may not perfectly reflect gene expression in the proximal colon, they were the best available data sources for this analysis.

- Page 11, lines 337-338: “In line with previous studies, we found that the genetic and dietary factors shape gut microbiome and BA composition and the response to HFD challenge.” In short: “...dietary factors shape... response to HFD challenge”. High-fat already represents a dietary factor. Hence, this statement does not seem to be appropriately formulated. Or are the authors referring to dietary factors other than the high-fat content in this sentence?

We acknowledge that this sentence was not well formatted and have now corrected it in the revised manuscript (Page 11, Lines 319-321).

References:

1. Williams, E. G. *et al.* Systems proteomics of liver mitochondria function. *Science* **352**, aad0189 (2016).
2. Wu, Y. *et al.* Multilayered genetic and omics dissection of mitochondrial activity in a mouse reference population. *Cell* **158**, 1415–1430 (2014).
3. Li, X. *et al.* Genetic and dietary modulators of the inflammatory response in the gastrointestinal tract of the BXD mouse genetic reference population. *eLife* **12**, RP87569 (2023).
4. Jha, P. *et al.* Genetic Regulation of Plasma Lipid Species and Their Association with Metabolic Phenotypes. *Cell Syst.* **6**, 709-721.e6 (2018).
5. Jha, P. *et al.* Systems Analyses Reveal Physiological Roles and Genetic Regulators of Liver Lipid Species. *Cell Syst.* **6**, 722-733.e6 (2018).

6. Li, H. *et al.* Integrative systems analysis identifies genetic and dietary modulators of bile acid homeostasis. *Cell Metab.* **34**, 1594-1610.e4 (2022).
7. Nagarajan, A., Scoggin, K., Gupta, J., Threadgill, D. W. & Andrews-Polymenis, H. L. Using the collaborative cross to identify the role of host genetics in defining the murine gut microbiome. *Microbiome* **11**, 149 (2023).
8. Chen, L. *et al.* The long-term genetic stability and individual specificity of the human gut microbiome. *Cell* **184**, 2302-2315.e12 (2021).
9. Kemis, J. H. *et al.* Genetic determinants of gut microbiota composition and bile acid profiles in mice. *PLoS Genet.* **15**, e1008073 (2019).
10. Yan, Y. *et al.* Bacteroides uniformis-induced perturbations in colonic microbiota and bile acid levels inhibit TH17 differentiation and ameliorate colitis developments. *Npj Biofilms Microbiomes* **9**, 1–14 (2023).
11. Honda, A. *et al.* Regulation of bile acid metabolism in mouse models with hydrophobic bile acid composition. *J. Lipid Res.* **61**, 54–69 (2020).

General response to all reviewers:

We thank the reviewers for providing valuable comments and suggestions. Please find all revised figures and text (highlighted in yellow) in the revised manuscript.

Reviewer #3 (Remarks to the Author):

All my concerns have been addressed in a satisfactory manner and I have no further questions.

Thank you.

Reviewer #4 (Remarks to the Author):

Thank you for co-reviewing our manuscript.

Reviewer #5 (Remarks to the Author):

The authors have clearly improved the quality of the manuscript during the revision process. This reviewer is satisfied with the authors' responses and the new data provided.

Thank you.

Reviewer #6 (Remarks to the Author):

In this work the authors seek to characterise the effects of high-fat diet on the mouse microbiome, trying to understand if any genetic variants and genes are involved in influencing these effects. The authors then try to use human genetic datasets to validate their results and link them to human health.

As my expertise is mostly on human genetics I will focus my comments on this section and leave to the other reviewers the others.

Overall the human genetics part of the paper may make sense in design where the authors try to reproduce their mice results in human verifying if differences in expression of the identified genes in the colon may lead to differences in the microbiome, using publicly available summary data.

We thank you for your encouraging comments.

There are several issue with the whole procedure:

1) The MR analysis is not described at all. The only mention is that they used a PCA based method and a reference. No broad description of the method is present or why this was chosen amongst the dozen existing. The choice of genetic instruments is not described (which SNPs were used? how were they selected? What was the power of the selected instruments? etc. etc). These information are fundamental to evaluate the results.

We have updated the method section based on your suggestions. Further explanation has been provided to justify the use of the PCA-based approach, detailing what it accomplishes and why it was selected. The SNPs used as the basis for the MR analysis (as PC instrumental variables) were the significant *cis*-eQTLs obtained directly from GTEx (v8). No additional filtering was applied, other than requiring that the variants be present in the outcome GWAS. The complete list of instrumental variables is provided in the Supplemental information 2. Please find the updated method below or in the revised manuscript (Pages 16-17, Lines 516-539).

Mendelian randomization analysis

Summary statistics-based Mendelian randomization was used to estimate the causal effects of the tissue-specific expression of candidate genes on various outcomes. Significant *cis*-eQTLs of candidate genes in the sigmoid colon and transverse colon were selected and their effect sizes were obtained from the GTEx Portal¹ on 2023-03-28 (v8, <https://www.gtexportal.org/home/datasets>). GWAS summary statistics for phenotypes of interest, namely the bacterial genera *Bacteroides* and *Turicibacter* (based on MiBioGen database) and their species *B. uniformis* and *T. sanguinis*, were obtained from the IEU OpenGWAS project^{2,3}, using the *ieugwasr* package (version 0.1.5). In most cases, the summary statistics of multiple significant *cis*-eQTLs were also available in the outcome GWAS, enabling their use as instrumental variables (IVs). Because only *cis*-eQTLs were selected, the IVs present a high level of linkage disequilibrium (LD) between each other and LD pruning would leave very few independent IVs, which would make the effect estimates less robust and liable to change based on which IVs are selected. Instead, we used an existing PCA-based approach⁴ which combines IVs using LD-based PCA, yielding PC IVs which are independent by construction. The LD structure was estimated based on the genotype files from GTEx and extracted using PLINK v1.90b6.21⁵. We selected top PC IVs to explain 99% of the genetic variance and used the inverse-variance weighted approach⁴ to estimate the causal effect. For results where the causal effect estimate was significant, we also performed a sensitivity test with a leave-one-out approach, removing one PC and estimating the causal effects using the remaining PCs. The resulting estimated causal effects were tested for heterogeneity using Cochran's Q test. The list of IVs is also provided (**Supplemental information 2**). In cases where only a single IV was available in both GTEx and the outcome GWAS (or if a single PC was retained), we used the Wald ratio method in TwoSampleMR⁶.

Furthermore no sensitivity analysis was conducted to verify the sturdiness of the results. So although I think the idea is potentially good there needs the analysis needs to be conducted and reported in much more thoroughly and with many more details.

We have added a leave-one-out approach as a sensitivity analysis, essentially leaving out one PC and re-estimating the causal effect. The resulting estimates for the causal effect of *PTGRI* expression in the colon on *Turicibacter* abundance were very consistent, with no significant heterogeneity observed (Cochran's Q test $p = 0.8$). The method of sensitivity analysis was also included in the Method section, please find it above or in the revised manuscript (Pages 17, Lines 533-537). We now also added the result of sensitivity analysis in the result section (Pages 9-10, Lines 262-265), which can be also found below.

For genes under gMxB1 QTL, MR suggested that the gene expression of prostaglandin reductase 1 (*PTGRI*) in human sigmoid colon increases fecal *Turicibacter* abundance (Beta = 0.09 and BH-adjusted P value = 0.039, Fig. 6b) with no significant heterogeneity observed (Cochran's Q test p = 0.8) in the sensitivity analysis.

This could be complemented with colocalization analysis which is quite standard now a days. This could be applied also to the disease outcomes to show potential mediation of the microbiome on disease.

We agree with your suggestion. We have now performed colocalization analysis using the coloc R package (version 5.2.3). The posterior probabilities (PP, hypothesis H4) for sharing a common causal variant between *PTGRI* or *PTPRD* eQTLs and *Turicibacter* GWAS were 0.018 and 0.15, respectively. The PP (H4) for *GABRB3* eQTLs and *Bacteroides* GWAS was 0.12. These results indicate that there is insufficient evidence to suggest shared causal variants affecting both traits. A likely reason for this, despite a significant MR result, is that these traits (the genes or the microbiome) lack strong genetic evidence at this locus due to, among others, small sample sizes. In particular, microbiome GWAS face additional challenges of the strong effect of the environment. The MiBioGen consortium only reaches a total sample size of 18,340 by integrating 24 cohorts from different populations and meta-analysis. Therefore, colocalization, which typically requires stronger evidence to yield significant results, is considerably underpowered in this situation.

As suggested, we also performed colocalization analysis between microbiome and metabolic trait GWAS signals within the regions of our candidate genes to assess whether the microbiome may mediate changes in related metabolic phenotypes. As mentioned above, the GWAS signals for each microbiome trait were relatively weak; therefore, we did not find strong evidence that the microbiome and metabolic traits share common causal variants. However, we did find suggestive evidence that shared variants within *PTPRD* may affect *Turicibacter* abundance and traits such as waist circumference and blood pressure, while variants within *GABRB3* may affect *Bacteroides* abundance and hip circumference (Figure R1). These findings might indicate a possible mediating role of the microbiome in metabolic health.

Figure R1. Scatter plot of posterior probabilities (PP, H4) for shared causal variants between microbiome and metabolic trait GWAS signals in each corresponding genomic region.

Given that the results of the colocalization analyses were inconclusive, we opted not to include them in the revised manuscript and show them only in the rebuttal.

2) Similarly the choices for the look up in GWAS is not clear as they looked specifically only to WES where imputed data exists and also WGS. Why did the authros focus only on coding variants given that their results showed an effect in differences of expression which is likely due to non-coding variants? Generally coding variants are analysed through aggregation tests

(ie Burden/SKAT) as those with high impact will have very low frequency. For UKB there is even a public database with previously reported analyses in UKB (GeneBass) which the authors could look for evidence of their results.

Thank you for your suggestion. We have now checked the UKBB GeneBass database to retrieve the burden test results of our three candidate genes. Even though we did not identify significant associations between the group of predicted loss of function or missense genetic variants within these genes and the metabolic traits (all Benjamini–Hochberg-adjusted $P > 0.05$), variants within *PTGR1* showed a suggestive association with heart failure, while those within *PTPRD* were suggestively associated with whole body fat mass, fat percentage, and alanine aminotransferase levels (Figure R2).

Figure R2. Line plot showing the associated metabolic phenotypic traits of genetic variants within our candidates in the burden test analysis.

In addition, we have now checked the GWAS summary statistic derived from UKBB whole genome sequencing (WGS)⁷. Genetic variants within our three candidate genes showed consistent suggestive or significant associations with metabolic traits, similar to the results from WES. These updated results are now presented in Figure 6e–g (page 25), while the panels displaying the GWAS results from UK Biobank whole-exome sequencing (WES) have been moved to Figure S4a–c (page 29).

Although there could be additional non-coding variants outside of the transcript (e.g., upstream of the transcription start site), we decided to focus on those within the transcript itself, since it is likely that if the gene itself is involved, it would have variants within the transcript that would be associated as well. This helps reduce the search space and restricts associations to variants most likely to be acting through our gene of interest.

So overall I think that the approach and results are interesting but that the analysis is still not deep and sturdy enough to answer to their scientific question.

Thank you, we think the new added data (sensitivity analysis and GWAS result derived from UKBB WGS) have now strengthened the study.

References:

1. GTEx Consortium. The Genotype-Tissue Expression (GTEx) project. *Nat. Genet.* **45**, 580–585 (2013).
2. Kurilshikov, A. *et al.* Large-scale association analyses identify host factors influencing human gut microbiome composition. *Nat. Genet.* **53**, 156–165 (2021).
3. Elsworth, B. *et al.* The MRC IEU OpenGWAS data infrastructure. 2020.08.10.244293 Preprint at <https://doi.org/10.1101/2020.08.10.244293> (2020).
4. Burgess, S., Zuber, V., Valdes-Marquez, E., Sun, B. B. & Hopewell, J. C. Mendelian randomization with fine-mapped genetic data: Choosing from large numbers of correlated instrumental variables. *Genet. Epidemiol.* **41**, 714–725 (2017).
5. Purcell, S. *et al.* PLINK: A Tool Set for Whole-Genome Association and Population-Based Linkage Analyses. *Am. J. Hum. Genet.* **81**, 559–575 (2007).

6. Hemani, G., Tilling, K. & Smith, G. D. Orienting the causal relationship between imprecisely measured traits using GWAS summary data. *PLoS Genet.* **13**, e1007081 (2017).
7. Carss, K. *et al.* Whole-genome sequencing of 490,640 UK Biobank participants. *Nature* **645**, 692–701 (2025).